# ScaleLong: Towards More Stable Training of Diffusion Model via Scaling Network Long Skip Connection

Zhongzhan Huang[1,2]    Pan Zhou[2*]    Shuicheng Yan[2]    Liang Lin[1*]

[1]Sun Yat-Sen University, [2]Sea AI Lab

huangzhzh23@mail2.sysu.edu.cn; zhoupan@sea.com;
shuicheng.yan@gmail.com; linliang@ieee.org

## Abstract

In diffusion models, UNet is the most popular network backbone, since its long skip connects (LSCs) to connect distant network blocks can aggregate long-distant information and alleviate vanishing gradient. Unfortunately, UNet often suffers from unstable training in diffusion models which can be alleviated by scaling its LSC coefficients smaller. However, theoretical understandings of the instability of UNet in diffusion models and also the performance improvement of LSC scaling remain absent yet. To solve this issue, we theoretically show that the coefficients of LSCs in UNet have big effects on the stableness of the forward and backward propagation and robustness of UNet. Specifically, the hidden feature and gradient of UNet at any layer can oscillate and their oscillation ranges are actually large which explains the instability of UNet training. Moreover, UNet is also provably sensitive to perturbed input, and predicts an output distant from the desired output, yielding oscillatory loss and thus oscillatory gradient. Besides, we also observe the theoretical benefits of the LSC coefficient scaling of UNet in the stableness of hidden features and gradient and also robustness. Finally, inspired by our theory, we propose an effective coefficient scaling framework ScaleLong that scales the coefficients of LSC in UNet and better improve the training stability of UNet. Experimental results on four famous datasets show that our methods are superior to stabilize training, and yield about $1.5\times$ training acceleration on different diffusion models with UNet or UViT backbones. Click here for Code.

## 1   Introduction

Recently, diffusion models (DMs) [1–9] have become the most successful generative models because of their superiority on modeling realistic data distributions. The methodology of DMs includes a forward diffusion process and a corresponding reverse diffusion process. For forward process, it progressively injects Gaussian noise into a realistic sample until the sample becomes a Gaussian noise, while the reverse process trains a neural network to denoise the injected noise at each step for gradually mapping the Gaussian noise into the vanilla sample. By decomposing the complex generative task into

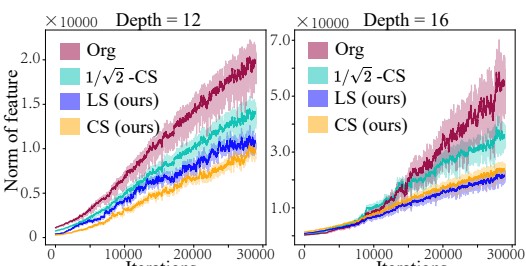

Figure 1: The feature oscillation in UNet.

a sequential application of denoising small noise, DMs achieve much better synthesis performance than other generative models, e.g., generative adversarial networks [10] and variational autoencoders [11], on image [12–15], 3D [16, 17] and video [18–21] data and beyond.

---

*Corresponding author.

37th Conference on Neural Information Processing Systems (NeurIPS 2023).

**Motivation.** Since the reverse diffusion process in DMs essentially addresses a denoising task, most DMs follow previous image denoising and restoration works [22–26] to employ UNet as their de-facto backbone. This is because UNet uses long skip connects (LSCs) to connect the distant and symmetrical network blocks in a residual network, which helps long-distant information aggregation and alleviates the vanishing gradient issue, yielding superior image denoising and restoration performance. However, in DMs, its reverse process uses a shared UNet to predict the injected noise at any step, even though the input noisy data have time-varied distribution due to the progressively injected noise. This inevitably leads to the training difficulty and instability of UNet in DMs. Indeed, as shown in Fig. 1, the output of hidden layers in UNet, especially for deep UNet, oscillates rapidly along with training iterations. This feature oscillation also indicates that there are some layers whose parameters suffer from oscillations which often impairs the parameter learning speed. Moreover, the unstable hidden features also result in an unstable input for subsequent layers and greatly increase their learning difficulty. Some works [27–31] empirically find that $1/\sqrt{2}$-constant scaling ($1/\sqrt{2}$-CS), that scales the coefficients of LSC (i.e. $\{\kappa_i\}$ in Fig. 2) from one used in UNet to $1/\sqrt{2}$, alleviates the oscillations to some extent as shown in Fig. 1, which, however, lacks intuitive and theoretical explanation and still suffers from feature and gradient oscillations. So it is unclear yet 1) why UNet in DMs is unstable and also 2) why scaling the coefficients of LSC in UNet helps stabilize training, which hinders the designing new and more advanced DMs in a principle way.

**Contribution.** In this work, we address the above two funda-mental questions and contribute to deriving some new results and alternatives for UNet. Particularly, we theoretically analyze the above training instability of UNet in DMs, and identify the key role of the LSC coefficients in UNet for its unstable training, interpreting the stabilizing effect and limitations of the $1/\sqrt{2}$-scaling technique. Inspired by our theory, we propose the framework ScaleLong including two novel-yet-effective coefficient scaling methods which scale the coefficients of LSC in UNet and better improve the training stability of UNet. Our main contributions are highlighted below.

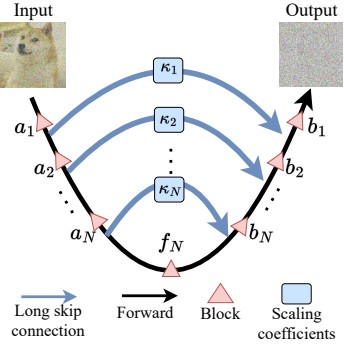

Figure 2: The diagram of UNet.

Our first contribution is proving that the coefficients of LSCs in UNet have big effects on the stableness of the forward and backward propagation and robustness of UNet. Formally, *for forward propagation*, we show that the norm of the hidden feature at any layer can be lower- and also upper-bounded, and the oscillation range between lower and upper bounds is of the order $\mathcal{O}\big(m\sum_{i=1}^{N}\kappa_j^2\big)$, where $\{\kappa_i\}$ denotes the scaling coefficients of $N$ LSCs and the input dimension $m$ of UNet is often of hundreds. Since standard UNet sets $\kappa_i = 1$ $(\forall i)$, the above oscillation range becomes $\mathcal{O}\big(mN\big)$ and is large, which partly explains the oscillation and instability of hidden features as observed in Fig. 1. The $1/\sqrt{2}$-CS technique can slightly reduce the oscillation range, and thus helps stabilize UNet. Similarly, *for backward propagation*, the gradient magnitude in UNet is upper bounded by $\mathcal{O}\big(m\sum_{i=1}^{N}\kappa_i^2\big)$. In standard UNet, this bound becomes a large bound $\mathcal{O}\big(mN\big)$, and yields the possible big incorrect parameter update, impairing the parameter learning speed during training. This also explains the big oscillation of hidden features, and reveals the stabilizing effects of the $1/\sqrt{2}$-CS technique. Furthermore, *for robustness* that measures the prediction change when adding a small perturbation into an input, we show robustness bound of UNet is $\mathcal{O}\big(\sum_{i=1}^{N}\kappa_i M_0^i\big)$ with a model-related constant $M_0 > 1$. This result also implies a big robustness bound of standard UNet, and thus shows the sensitivity of UNet to the noise which can give a big incorrect gradient and explain the unstable training. It also shows that $1/\sqrt{2}$-CS technique improves the robustness.

Inspired by our theory, we further propose a novel framework ScaleLong including two scaling methods to adjust the coefficients of the LSCs in UNet for stabilizing the training: 1) constant scaling method (**CS** for short) and 2) learnable scaling method (**LS**). CS sets the coefficients $\{\kappa_i\}$ as a serir of exponentially-decaying constants, i.e., $\kappa_i = \kappa^{i-1}$ with a contant $\kappa \in (0, 1]$. As a result, CS greatly stabilizes the UNettraining by largely reducing the robustness error bound from $\mathcal{O}(M_0^N)$ to $\mathcal{O}\big(M_0(\kappa M_0)^{N-1}\big)$, which is also $\mathcal{O}(\kappa^{N-2})$ times smaller than the bound $\mathcal{O}(\kappa M_0^N)$ of the universal scaling method, i.e., $\kappa_i = \kappa$, like $1/\sqrt{2}$-CS technique. Meanwhile, CS can ensure the information transmission of LSC without degrading into a feedforward network. Similarly, the oscillation range

of hidden features and also gradient magnitude can also be controlled by our CS. For LS, it uses a tiny shared network for all LSCs to predict the scaling coefficients for each LSC. In this way, LS is more flexible and adaptive than the CS method, as it can learn the scaling coefficients according to the training data, and can also adjust the scaling coefficients along with training iterations which could benefit the training in terms of stability and convergence speed. Fig. 1 shows the effects of CS and LS in stabilizing the UNet training in DMs.

Extensive experiments on CIFAR10 [32], CelebA [33], ImageNet [34], and COCO [35], show the effectiveness of our CS and LS methods in enhancing training stability and thus accelerating the learning speed by at least $1.5\times$ in most cases across several DMs, e.g. UNet and UViT networks under the unconditional [3–5], class-conditional [36, 37] and text-to-image [38–42] settings.

## 2  Preliminaries and other related works

**Diffusion Model (DM)**. DDPM-alike DMs [1–6, 8, 9, 43] generates a sequence of noisy samples $\{\mathbf{x}_i\}_{i=1}^{T}$ by repeatedly adding Gaussian noise to a sample $\mathbf{x}_0$ until attatining $\mathbf{x}_T \sim \mathcal{N}(\mathbf{0}, \mathbf{I})$. This noise injection process, a.k.a. the forward process, can be formalized as a Markov chain $q(\mathbf{x}_{1:T}|\mathbf{x}_0) = \prod_{t=1}^{T} q(\mathbf{x}_t|\mathbf{x}_{t-1})$, where $q(\mathbf{x}_t|\mathbf{x}_{t-1}) = \mathcal{N}(\mathbf{x}_t|\sqrt{\alpha_t}\mathbf{x}_{t-1}, \beta_t\mathbf{I})$, $\alpha_t$ and $\beta_t$ depend on $t$ and satisfy $\alpha_t + \beta_t = 1$. By using the properties of Gaussian distribution, the forward process can be written as

$$q(\mathbf{x}_t|\mathbf{x}_0) = \mathcal{N}(\mathbf{x}_t; \sqrt{\bar{\alpha}_t}\mathbf{x}_0, (1 - \bar{\alpha}_t)\mathbf{I}), \tag{1}$$

where $\bar{\alpha}_t = \prod_{i=1}^{t} \alpha_i$. Next, one can sample $\mathbf{x}_t = \sqrt{\bar{\alpha}_t}\mathbf{x}_0 + \sqrt{1 - \bar{\alpha}_t}\epsilon_t$, where $\epsilon_t \sim \mathcal{N}(\mathbf{0}, \mathbf{I})$. Then DM adopts a neural network $\hat{\epsilon}_\theta(\cdot, t)$ to invert the forward process and predict the noise $\epsilon_t$ added at each time step. This process is to recover data $\mathbf{x}_0$ from a Gaussian noise by minimizing the loss

$$\ell_{\text{simple}}^{t}(\theta) = \mathbb{E}\|\epsilon_t - \hat{\epsilon}_\theta(\sqrt{\bar{\alpha}_t}\mathbf{x}_0 + \sqrt{1 - \bar{\alpha}_t}\epsilon_t, t)\|_2^2. \tag{2}$$

**UNet-alike Network**. Since the network $\hat{\epsilon}_\theta(\cdot, t)$ in DMs predicts the noise to denoise, it plays a similar role of UNet-alike network widely in image restoration tasks, e.g. image de-raining, image denoising [22–26, 44, 45]. This inspires DMs to use UNet-alike network in (3) that uses LSCs to connect distant parts of the network for long-range information transmission and aggregation

$$\mathbf{UNet}(x) = f_0(x), \quad f_i(x) = b_{i+1} \circ [\kappa_{i+1} \cdot a_{i+1} \circ x + f_{i+1}(a_{i+1} \circ x)], \ 0 \le i \le N-1 \tag{3}$$

where $x \in \mathbb{R}^m$ denotes the input, $a_i$ and $b_i$ $(i \ge 1)$ are the trainable parameter of the $i$-th block, $\kappa_i > 0$ $(i \ge 1)$ are the scaling coefficients and are set to 1 in standard UNet. $f_N$ is the middle block of UNet. For the vector operation $\circ$, it can be designed to implement different networks.

W.l.o.g, in this paper, we consider $\circ$ as matrix multiplication, and set the $i$-th block as a stacked network [46, 47] which is implemented as $a_i \circ x = \mathbf{W}_l^{a_i}\phi(\mathbf{W}_{l-1}^{a_i}...\phi(\mathbf{W}_1^{a_i}x))$ with a ReLU activation function $\phi$ and learnable matrices $\mathbf{W}_j^{a_i} \in \mathbb{R}^{m \times m}$ $(j \ge 1)$. Moreover, let $f_N$ also have the same architecture, i.e. $f_N(x) = \mathbf{W}_l^{f_i}\phi(\mathbf{W}_{l-1}^{f_i}...\phi(\mathbf{W}_1^{f_i}x))$. Following [36], the above UNet has absorbed the fusion operation of the feature $a_i \circ x$ and $f_i(a_i \circ x)$ into UNet backbone. So for simplicity, we analyze the UNet in (3). Moreover, as there are many variants of UNet, e.g. transformer UNet, it is hard to provide a unified analysis for all of them. Thus, here we only analyze the most widely used UNet in (3), and leave the analysis of other UNet variants in future work.

**Other related works.** Previous works [46, 48–55] mainly studied classification task and observed performance improvement of scaling block output instead of skip connections in ResNet and Transformer, i.e. $\mathbf{x}_{i+1} = \mathbf{x}_i + \kappa_i f_i(\mathbf{x}_i)$ where $f_i(\mathbf{x}_i)$ is the $i$-th block. In contrast, we study UNet for DMs which uses long skip connections (LSC) instead of short skip connections, and stabilize UNet training via scaling LSC via Eq. (3) which is superior to scaling block output as shown in Section 5.

## 3  Stability Analysis on UNet

As shown in Section 1, standard U-Net ($\kappa_i = 1, \forall i$) often suffers from the unstable training issue in diffusion models (DMs), while scaling the long skip connection (LSC) coefficients $\kappa_i(\forall i)$ in UNet can help to stabilize the training. However, theoretical understandings of the instability of U-Net and also the effects of scaling LSC coeffeicents remain absent yet, hindering the development of new and more advanced DMs in a principle way. To address this problem, in this section, we will theoretically and comprehensively analyze 1) why UNet in DMs is unstable and also 2) why scaling

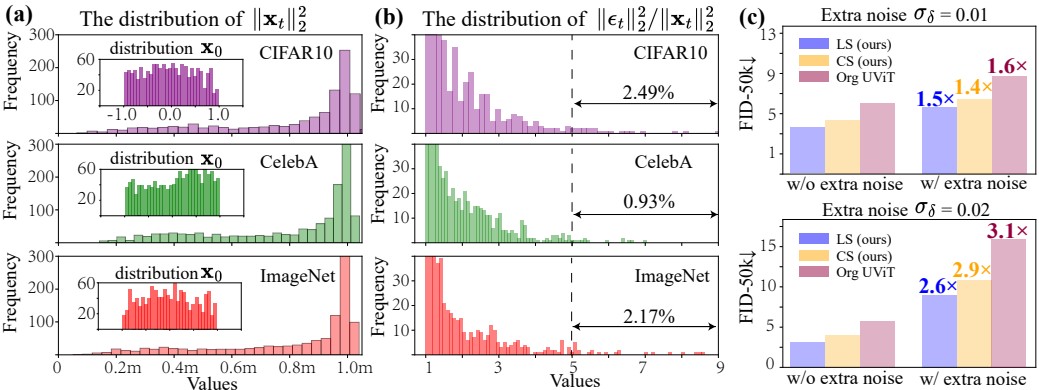

Figure 3: Distribution of $\mathbf{x}_0$, $\|\mathbf{x}_t\|_2^2$ and $\|\epsilon_t\|_2^2/\|\mathbf{x}_t\|_2^2$, and effect of extra noise to performance.

the coeffeicents of LSC in UNet helps stablize training. To this end, in the following, we will analyze the stableness of the forward and backward propagation of UNet, and investigate the robustness of UNet to the noisy input. All these analyses not only deepen the understanding of the instability of UNet, but also inspire an effective solution for a more stable UNet as introduced in Section 4.

## 3.1 Stability of Forward Propagation

Here we study the stableness of the hidden features of UNet. This is very important, since for a certain layer, if its features vary significantly along with the training iterations, it means there is at least a layer whose parameters suffer from oscillations which often impairs the parameter learning speed. Moreover, the unstable hidden features also result in an unstable input for subsequent layers and greatly increase their learning difficulty, since a network cannot easily learn an unstable distribution, e.g. Internal Covariate Shift [56, 57], which is also shown in many works [58–60]. In the following, we theoretically characterize the hidden feature output by bounding its norm.

**Theorem 3.1.** *Assume that all learnable matrices of UNet in Eq. (3) are independently initialized as Kaiming's initialization, i.e., $\mathcal{N}(0, \frac{2}{m})$. Then for any $\rho \in (0, 1]$, by minimizing the training loss Eq. (2) of DMs, with probability at least $1 - \mathcal{O}(N)\exp[-\Omega(m\rho^2)]$, we have*

$$(1 - \rho)^2 \left[ c_1 \|\mathbf{x}_t\|_2^2 \cdot \sum_{j=i}^{N} \kappa_j^2 + c_2 \right] \lesssim \|h_i\|_2^2 \lesssim (1 + \rho)^2 \left[ c_1 \|\mathbf{x}_t\|_2^2 \cdot \sum_{j=i}^{N} \kappa_j^2 + c_2 \right], \quad (4)$$

*where the hidden feature $h_i$ is the output of $f_i$ defined in Eq. (3), $\mathbf{x}_t = \sqrt{\bar{\alpha}_t}\mathbf{x}_0 + \sqrt{1 - \bar{\alpha}_t}\epsilon_t$ is the input of UNet; $c_1$ and $c_2$ are two constants; $\kappa_i$ is the scaling coefficient of the $i$-th LSC in UNet.*

Theorem 3.1 reveals that with high probability, the norm of hidden features $h_i$ of UNet can be both lower- and also upper-bounded, and its biggest oscillation range is of the order $\mathcal{O}\left(\|\mathbf{x}_t\|_2^2 \sum_{j=i}^{N} \kappa_j^2\right)$ which is decided by the scaling coefficients $\{\kappa_i\}$ and the UNet input $\mathbf{x}_t$. So when $\|\mathbf{x}_t\|_2^2$ or $\sum_{i=1}^{N} \kappa_i^2$ is large, the above oscillation range is large which allows hidden feature $h_i$ to oscillate largely as shown in Fig. 1, leading to the instability of the forward processing in UNet.

In fact, the following Lemma 3.2 shows that $\|\mathbf{x}_t\|_2^2$ is around feature dimension $m$ in DDPM.

**Lemma 3.2.** *For $\mathbf{x}_t = \sqrt{\bar{\alpha}_t}\mathbf{x}_0 + \sqrt{1 - \bar{\alpha}_t}\epsilon_t$ defined in Eq. (1) as a input of UNet, $\epsilon_t \sim \mathcal{N}(0, \mathbf{I})$, if $\mathbf{x}_0$ follow the uniform distribution $U[-1, 1]$, then we have*

$$\mathbb{E}\|\mathbf{x}_t\|_2^2 = (1 - 2\mathbb{E}_t\bar{\alpha}_t/3)m = \mathcal{O}(m). \quad (5)$$

Indeed, the conditions and conclusions of Lemma 3.2 hold universally in commonly used datasets, such as CIFAR10, CelebA, and ImageNet, which can be empirically valid in Fig. 3 (a). In this way, in standard UNet whose scaling coefficients satisfy $\kappa_i = 1$, the oscillation range of hidden feature $\|h_i\|_2^2$ becomes a very large bound $\mathcal{O}(Nm)$, where $N$ denotes the number of LSCs. Accordingly, UNet suffers from the unstable forward process caused by the possible big oscillation of the hidden feature $h_i$ which accords with the observations in Fig. 1. To relieve the instability issue, one possible

solution is to scale the coefficients $\{\kappa_i\}$ of LSCs, which could reduce the oscillation range of hidden feature $\|h_i\|_2^2$ and yield more stable hidden features $h_i$ to benefit the subsequent layers. This is also one reason why $1/\sqrt{2}$-scaling technique [27–31] can help stabilize the UNet. However, it is also important to note that these coefficients cannot be too small, as they could degenerate a network into a feedforward network and negatively affect the network representation learning ability.

## 3.2 Stability of Backward Propagation

As mentioned in Section 3.1, the reason behind the instability of UNet forward process in DMs is the parameter oscillation. Here we go a step further to study why these UNet parameters oscillate. Since parameters are updated by the gradient given in the backward propagation process, we analyze the gradient of UNet in DMs. A stable gradient with proper magnitude can update model parameters smoothly and also effectively, and yields more stable training. On the contrary, if gradients oscillate largely as shown in Fig. 1, they greatly affect the parameter learning and lead to the unstable hidden features which further increases the learning difficulty of subsequent layers. In the following, we analyze the gradients of UNet, and investigate the effects of LSC coefficient scaling on gradients.

For brivity, we use $\mathbf{W}$ to denote all the learnable matrices of the $i$-th block (see Eq. (3)), i.e., $\mathbf{W} = [\mathbf{W}_l^{a_1}, \mathbf{W}_{l-1}^{a_1}, ..., \mathbf{W}_1^{a_1}, \mathbf{W}_l^{a_2}..., \mathbf{W}_2^{b_1}, \mathbf{W}_1^{b_1}]$. Assume the training loss is $\frac{1}{n}\sum_{s=1}^{n}\ell_s(\mathbf{W})$, where $\ell_s(\mathbf{W})$ denotes the training loss of the $s$-th sample among the $n$ training samples. Next, we analyze the gradient $\nabla\ell(\mathbf{W})$ of each training sample, and summarize the main results in Theorem 3.3, in which we omit the subscript $s$ of $\nabla\ell_s(\mathbf{W})$ for brevity. See the proof in Appendix.

**Theorem 3.3.** *Assume that all learnable matrices of UNet in Eq. (3) are independently initialized as Kaiming's initialization, i.e., $\mathcal{N}(0, \frac{2}{m})$. Then for any $\rho \in (0, 1]$, with probability at least $1 - \mathcal{O}(nN)\exp[-\Omega(m)]$, for a sample $\mathbf{x}_t$ in training set, we have*

$$\|\nabla_{\mathbf{W}_p^q}\ell_s(\mathbf{W})\|_2^2 \lesssim \mathcal{O}\left(\ell(\mathbf{W}) \cdot \|\mathbf{x}_t\|_2^2 \cdot \sum_{j=i}^{N}\kappa_j^2 + c_3\right), \quad (p \in \{1, 2, ..., l\}, q \in \{a_i, b_i\}), \quad (6)$$

*where $\mathbf{x}_t = \sqrt{\bar{\alpha}_t}\mathbf{x}_0 + \sqrt{1-\bar{\alpha}_t}\epsilon_t$ denotes the noisy sample of the $s$-th sample, $\epsilon_t \sim \mathcal{N}(\mathbf{0}, \mathbf{I})$, $N$ is the number of LSCs, $c_3$ is a small constant.*

Theorem 3.3 reveals that for any training sample in the training dataset, the gradient of any trainable parameter $\mathbf{W}_p^q$ in UNet can be bounded by $\mathcal{O}(\ell(\mathbf{W}) \cdot \|\mathbf{x}_t\|_2^2 \sum_{j=i}^{N}\kappa_j^2)$ (up to the constant term). Since Lemma 3.2 shows $\|\mathbf{x}_t\|_2^2$ is at the order of $\mathcal{O}(m)$ in expectation, the bound of gradient norm becomes $\mathcal{O}(m\ell_s(\mathbf{W})\sum_{j=i}^{N}\kappa_j^2)$. Consider the feature dimension $m$ is often hundreds and the loss $\ell(\mathbf{W})$ would be large at the beginning of the training phase, if the number $N$ of LSCs is relatively large, the gradient bound in standard UNet ($\kappa_i = 1 \,\forall i$) would become $\mathcal{O}(mN\ell_s(\mathbf{W}))$ and is large. This means that UNet has the risk of instability gradient during training which is actually observed in Fig. 1. To address this issue, similar to the analysis of Theorem 3.3, we can appropriately scale the coefficients $\{\kappa_i\}$ to ensure that the model has enough representational power while preventing too big a gradient and avoid unstable UNet parameter updates. This also explains why the $1/\sqrt{2}$-scaling technique can improve the stability of UNet in practice.

## 3.3 Robustness On Noisy Input

In Section 3.1 and 3.2, we have revealed the reasons of instability of UNet and also the benign effects of scaling coefficients of LSCs in UNet during DM training. Here we will further discuss the stability of UNet in terms of the robustness on noisy input. A network is said to be robust if its output does not change much when adding a small noise perturbation to its input. This robustness is important for stable training of UNet in DMs. As shown in Section 2, DMs aim at using UNet to predict the noise $\epsilon_t \sim \mathcal{N}(0, \mathbf{I})$ in the noisy sample for denoising at each time step $t$. However, in practice, additional unknown noise caused by random minibatch, data collection, and preprocessing strategies is inevitably introduced into noise $\epsilon_t$ during the training process, and greatly affects the training. It is because if a network is sensitive to noisy input, its output would be distant from the desired output, and yields oscillatory loss which leads to oscillatory gradient and parameters. In contrast, a robust network would have stable output, loss and gradient which boosts the parameter learning speed.

Here we use a practical example to show the importance of the robustness of UNet in DMs. According to Eq. (2), DM aims to predict noise $\epsilon_t$ from the UNet input $\mathbf{x}_t \sim \mathcal{N}(\mathbf{x}_t; \sqrt{\bar{\alpha}_t}\mathbf{x}_0, (1-\bar{\alpha}_t)\mathbf{I})$.

Assume an extra Gaussian noise $\epsilon_\delta \sim \mathcal{N}(0, \sigma_\delta^2 \mathbf{I})$ is injected into $\mathbf{x}_t$ which yields a new input $\mathbf{x}_t' \sim \mathcal{N}(\mathbf{x}_t; \sqrt{\bar{\alpha}_t}\mathbf{x}_0, [(1 - \bar{\alpha}_t) + \sigma_\delta^2]\mathbf{I})$. Accordingly, if the variance $\sigma_\delta^2$ of extra noise is large, it can dominate the noise $\epsilon_t$ in $\mathbf{x}_t'$, and thus hinders predicting the desired noise $\epsilon_t$. In practice, the variance of this extra noise could vary along training iterations, further exacerbating the instability of UNet training. Indeed, Fig. 3 (c) shows that the extra noise $\epsilon_\delta$ with different $\sigma_\delta^2$ can significantly affect the performance of standard UNet for DMs, and the scaled LSCs can alleviate the impact of extra noise to some extent. We then present the following theorem to quantitatively analyze these observations.

**Theorem 3.4.** *For UNet in Eq. (3), assume $M_0 = \max\{\|b_i \circ a_i\|_2, 1 \le i \le N\}$ and $f_N$ is $L_0$-Lipschitz continuous. $c_0$ is a constant related to $M_0$ and $L_0$. Suppose $\mathbf{x}_t^{\epsilon_\delta}$ is an perturbated input of the vanilla input $\mathbf{x}_t$ with a small perturbation $\epsilon_\delta = \|\mathbf{x}_t^{\epsilon_\delta} - \mathbf{x}_t\|_2$. Then we have*

$$\|\mathbf{UNet}(\mathbf{x}_t^{\epsilon_\delta}) - \mathbf{UNet}(\mathbf{x}_t)\|_2 \le \epsilon_\delta \left[ \sum_{i=1}^{N} \kappa_i M_0^i + c_0 \right], \tag{7}$$

*where $\mathbf{x}_t = \sqrt{\bar{\alpha}_t}\mathbf{x}_0 + \sqrt{1 - \bar{\alpha}_t}\epsilon_t$, $\epsilon_t \sim \mathcal{N}(\mathbf{0}, \mathbf{I})$, $N$ is the number of the long skip connections.*

Theorem 3.4 shows that for a perturbation magnitude $\epsilon_\delta$, the robustness error bound of UNet in DMs is $\mathcal{O}(\epsilon_\delta(\sum_{i=1}^{N} \kappa_i M_0^i))$. For standard UNet ($\kappa_i = 1 \forall i$), this bound becomes a very large bound $\mathcal{O}(N M_0^N)$. This implies that standard UNet is sensitive to extra noise in the input, especially when LSC number $N$ is large ($M_0 \ge 1$ in practice, see Appendix). Hence, it is necessary to control the coefficients $\kappa_i$ of LSCs which accords with the observations in Fig. 3 (c). Therefore, setting appropriate scaling coefficients $\{\kappa_i\}$ can not only control the oscillation of hidden features and gradients during the forward and backward processes as described in Section 3.1 and Section 3.2, but also enhance the robustness of the model to input perturbations, thereby improving better stability for DMs as discussed at the beginning of Section 3.3.

# 4 Theory-inspired Scaling Methods For Long Skip Connections

In Section 3, we have theoretically shown that scaling the coefficients of LSCs in UNet can help to improve the training stability of UNet on DM generation tasks by analyzing the stability of hidden feature output and gradient, and the robustness of UNet on noisy input. These theoretical results can directly be used to explain the effectiveness of $1/\sqrt{2}$-scaling technique in [27–31] which scales all LSC coefficients $\kappa_i$ from one in UNet to a constant, and is called $1/\sqrt{2}$ ($1/\sqrt{2}$-CS for short). Here we will use these theoretical results to design a novel framework ScaleLong including two more effective scaling methods for DM stable training. 1) constant scaling method (CS for short) and 2) learnable scaling method (LS for short).

## 4.1 Constant Scaling Method

In Theorem 3.1 $\sim$ 3.4, we already show that adjusting the scaling coefficients $\{\kappa_i\}$ to smaller ones can effectively decrease the oscillation of hidden features, avoids too large gradient and also improves the robustness of U-Net to noisy input. Motivated by this, we propose an effective constant scaling method (CS) that exponentially scales the coefficients $\{\kappa_i\}$ of LSCs in UNet:

$$\mathbf{CS:} \quad \kappa_i = \kappa^{i-1}, \qquad (i = 1, 2, .., N, \quad \forall \kappa \in (0, 1]). \tag{8}$$

When $i = 1$, the scaling coefficient of the first long skip connection is 1, and thus ensures that at least one $\kappa_i$ is not too small to cause network degeneration.

This exponential scaling in CS method is more effective than the universal scaling methods which universally scales $\kappa_i = \kappa$, e.g. $\kappa = 1/\sqrt{2}$ in the $1/\sqrt{2}$-CS method, in terms of reducing the training oscillation of UNet. We first look at the robustness error bound in Theorem 3.4. For our CS, its bound is at the order of $\mathcal{O}(M_0(\kappa M_0)^{N-1})$ which is $\mathcal{O}(\kappa^{N-2})$ times smaller than the bound $\mathcal{O}(\kappa M_0^N)$ of the universal scaling method. This shows the superiority of our CS method. Moreover, from the analysis in Theorem 3.1 , our CS method can effectively compress the oscillation range of the hidden feature $h_i$ of the $i$-th layer to $\mathcal{O}(\kappa^{2i}m)$. Compared with the oscillation range $\mathcal{O}(Nm)$ of standard UNet, CS also greatly alleviates the oscillation since $\kappa^{2i} \ll N$. A similar analysis on Theorem 3.3 also shows that CS can effectively control the gradient magnitude, helping stabilize UNet training.

Now we discuss how to set the value of $\kappa$ in Eq. (8). To this end, we use the widely used DDPM framework on the benchmarks (CIFAR10, CelebA and ImageNet) to estimate $\kappa$. Firstly, Theorem

[3.1](#) reveals that the norm of UNet output is of order $\mathcal{O}(\|\mathbf{x}_t\|_2^2 \sum_{j=1}^{N} \kappa_j^2)$. Next, from the loss function of DMs, i.e., Eq. ([2](#)), the target output is a noise $\epsilon_t \sim \mathcal{N}(0, \mathbf{I})$. Therefore, we expect that $\mathcal{O}(\|\mathbf{x}_t\|_2^2 \sum_{j=1}^{N} \kappa_j^2) \approx \|\epsilon_t\|_2^2$, and we have $\sum_{j=1}^{N} \kappa_j^2 \approx \mathcal{O}(\|\epsilon_t\|_2^2 / \|\mathbf{x}_t\|_2^2)$. Moreover, Fig. [3](#) (b) shows the distribution of $\|\epsilon_t\|_2^2 / \|\mathbf{x}_t\|_2^2$ for the dataset considered, which is long-tailed and more than 97% of the values are smaller than 5. For this distribution, during the training, these long-tail samples result in a larger value of $\|\epsilon_t\|_2^2 / \|\mathbf{x}_t\|_2^2$. They may cause the output order $\mathcal{O}(\|\mathbf{x}_t\|_2^2 \sum_{j=1}^{N} \kappa_j^2)$ to deviate more from the $\|\epsilon_t\|_2^2$, leading to unstable loss and impacting training. Therefore, it is advisable to appropriately increase $\sum_{j=1}^{N} \kappa_j^2$ to address the influence of long-tail samples. [[61](#)–[65](#)]. So the value of $\sum_{j=1}^{N} \kappa_j^2$ should be between the mean values of $\|\epsilon_t\|_2^2 / \|\mathbf{x}_t\|_2^2$, i.e., about 1.22, and a rough upper bound of 5. Combining Eq. ([8](#)) and the settings of $N$ in UViT and UNet, we can estimate $\kappa \in [0.5, 0.95]$. In practice, the hyperparameter $\kappa$ can be adjusted around this range.

## 4.2   Learnable Scaling Method

In Section [4.1](#), an effective constant scaling (CS) method is derived from theory, and can improve the training stability of UNet in practice as shown in Section [5](#). Here, we provide an alternative solution, namely, the learnable scaling method (LS), which uses a tiny network to predict the scaling coefficients for each long skip connection. Accordingly, LS is more flexible and adaptive than the constant scaling method, since it can learn the coefficients according to the training data, and can also adjust the coefficients along with training iterations which could benefit the training in terms of stability and convergence speed, which will be empirically demonstrated in Section [5](#).

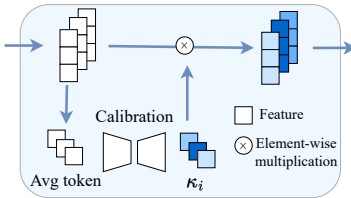

Figure 4: The diagram of LS.

As shown in Fig. [4](#), LS designs a calibration network usually shared by all long skip connections to predict scaling coefficients $\{\kappa_i\}$. Specifically, for the $i$-th long skip connection, its input feature is $x_i \in \mathbb{R}^{B \times N \times D}$, where $B$ denotes the batch size. For convolution-based UNet, $N$ and $D$ respectively denote the channel number and spatial dimension ($H \times W$) of the feature map; for transformer-based UViT [[36](#)], $N$ and $D$ respectively represent the token number and token dimension. See more discussions on input features of other network architectures in the Appendix. Accordingly, LS feeds $x_i \in \mathbb{R}^{B \times N \times D}$ into the tiny calibration network $\zeta_\phi$ parameterized by $\phi$ for prediction:

$$\textbf{LS:} \quad \kappa_i = \sigma(\zeta_\phi[\text{GAP}(x_i)]) \in \mathbb{R}^{B \times N \times 1}, 1 \leq i \leq N, \tag{9}$$

where GAP denotes the global average pooling that averages $x_i$ along the last dimension to obtain a feature map of size $B \times N \times 1$; $\sigma$ is a sigmoid activation function. After learning $\{\kappa_i\}$, LS uses $\kappa_i$ to scale the input $x_i$ via element-wise multiplication. In practice, network $\zeta_\phi$ is very small, and has only about 0.01M parameters in our all experiments which brings an ignorable cost compared with the cost of UNet but greatly improves the training stability and generation performance of UNet. In fact, since the parameter count of $\zeta_\phi$ itself is not substantial, to make LS more versatile for different network architecture, we can individually set a scaling module into each long skip connection. This configuration typically does not lead to performance degradation and can even result in improved performance, while maintaining the same inference speed as the original LS.

## 5   Experiment

In this section, we evaluate our methods by using UNet [[3](#), [4](#)] and also UViT [[36](#), [66](#), [67](#)] under the unconditional [[3](#)–[5](#)] , class-conditional [[36](#), [37](#)] and text-to-image [[38](#)–[42](#)] settings on several commonly used datasets, including CIFAR10 [[32](#)], CelebA [[33](#)], ImageNet [[34](#)], and MS-COCO [[35](#)]. We follow the settings of [[36](#)] and defer the specific implementation details to the Appendix.

**Training Stability.** Fig. [1](#) shows that our CS and LS methods can significantly alleviate the oscillation in UNet training under different DM settings, which is consistent with the analysis for controlling hidden features and gradients in Section [3](#). Though the $1/\sqrt{2}$-CS method can stabilize training to some extent, there are relatively large oscillations yet during training, particularly for deep networks. Additionally, Fig. [3](#) (c) shows that CS and LS can resist extra noise interference to some extent, further demonstrating the effectiveness of our proposed methods in improving training stability.

**Convergence Speed**. Fig. [5](#) shows the training efficiency of our CS and LS methods under different settings which is a direct result of stabilizing effects of our methods. Specifically, during training, for

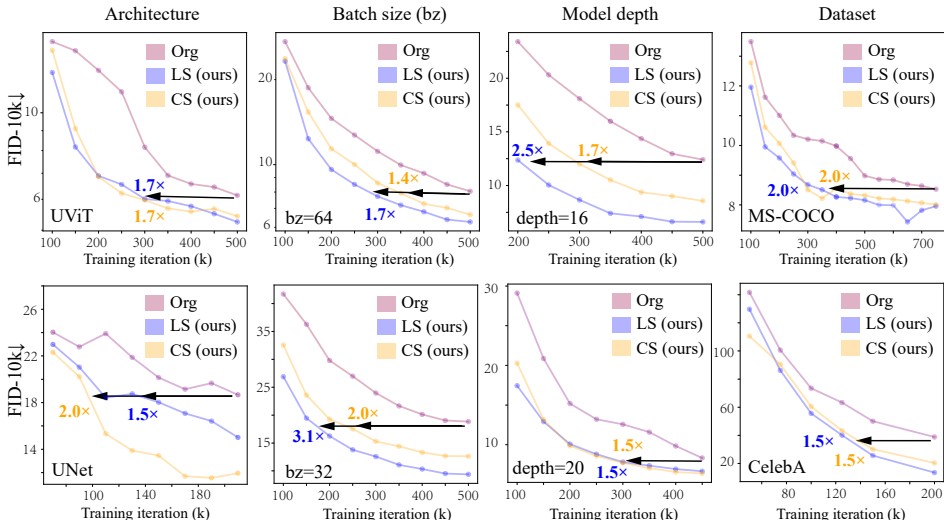

Figure 5: Training curve comparison under different training settings in which all experiments here adopt 12-layered UViT and a batch size 128 on CIFAR10 unless specified in the figures.

Table 1: Synthesis performance comparison under different diffusion model settings.

| Model (CIFAR10) | Type | #Param | FID↓ |
|---|---|---|---|
| EDM [68] | Uncondition | 56M | 1.97 |
| IDDPM [3] | Uncondition | 53M | 2.90 |
| DDPM++ cont. [27] | Uncondition | 62M | 2.55 |
| GenViT [70] | Uncondition | 11M | 20.20 |
| UNet [4] | Uncondition | 36M | 3.17 |
| UViT-S/2 [36] | Uncondition | 44M | 3.11 |
| UNet+1/√2-CS | Uncondition | 36M | 3.14 |
| UViT-S/2-1/√2-CS | Uncondition | 44M | 3.11 |
| UNet+CS (ours) | Uncondition | 36M | 3.05 |
| UNet+LS (ours) | Uncondition | 36M | 3.07 |
| UViT-S/2+CS (ours) | Uncondition | 44M | 2.98 |
| UViT-S/2+LS (ours) | Uncondition | 44M | 3.01 |

| Model (MS-COCO) | Type | #Param | FID↓ |
|---|---|---|---|
| VQ-Diffusion [41] | Text-to-Image | 370M | 19.75 |
| Friro [72] | Text-to-Image | 766M | 8.97 |
| UNet [36] | Text-to-Image | 260M | 7.32 |
| UNet+1/√2-CS | Text-to-Image | 260M | 7.19 |
| UViT-S/2+1/√2-CS | Text-to-Image | 252M | 5.88 |
| UViT-S/2 [36] | Text-to-Image | 252M | 5.95 |
| UViT-S/2 (Deep) [36] | Text-to-Image | 265M | 5.48 |
| UNet+CS (ours) | Text-to-Image | 260M | 7.04 |
| UNet+LS (ours) | Text-to-Image | 260M | 6.89 |
| UViT-S/2+CS (ours) | Text-to-Image | 252M | 5.77 |
| UViT-S/2+LS (ours) | Text-to-Image | 252M | 5.60 |

| Model (ImageNet 64) | Type | #Param | FID↓ |
|---|---|---|---|
| Glow [69] | - | - | 3.81 |
| IDDPM (small) [3] | Class-condition | 100M | 6.92 |
| IDDPM (large) [3] | Class-condition | 270M | 2.92 |
| ADM [71] | Class-condition | 296M | 2.07 |
| EDM [68] | Class-condition | 296M | 1.36 |
| UViT-M/4 [36] | Class-condition | 131M | 5.85 |
| UViT-L/4 [36] | Class-condition | 287M | 4.26 |
| UViT-M/4-1/√2-CS | Class-condition | 131M | 5.80 |
| UViT-L/4-1/√2-CS | Class-condition | 287M | 4.20 |
| UNet [3] | Class-condition | 144M | 3.77 |
| UNet-1/√2-CS | Class-condition | 144M | 3.66 |
| UNet+CS (ours) | Class-condition | 144M | 3.48 |
| UNet+LS (ours) | Class-condition | 144M | 3.39 |
| UViT-M/4+CS (ours) | Class-condition | 131M | 5.68 |
| UViT-M/4+LS (ours) | Class-condition | 131M | 5.75 |
| UViT-L/4+CS (ours) | Class-condition | 287M | 3.83 |
| UViT-L/4+LS (ours) | Class-condition | 287M | 4.08 |

| Model (CelebA) | Type | #Param | FID↓ |
|---|---|---|---|
| DDIM [73] | Uncondition | 79M | 3.26 |
| Soft Truncation [74] | Uncondition | 62M | 1.90 |
| UViT-S/2 [36] | Uncondition | 44M | 2.87 |
| UViT-S/2-1/√2-CS | Uncondition | 44M | 3.15 |
| UViT-S/2+CS (ours) | Uncondition | 44M | 2.78 |
| UViT-S/2+LS (ours) | Uncondition | 44M | 2.73 |

most cases, they consistently accelerate the learning speed by at least $1.5\times$ faster in terms of training steps across different datasets, batch sizes, network depths, and architectures. This greatly saves the actual training time of DMs. For example, when using a 32-sized batch, LS trained for 6.7 hours (200 k steps) achieves superior performance than standard UViT trained for 15.6 hours (500 k steps). All these results reveal the effectiveness of the long skip connection scaling for faster training of DMs.

**Performance Comparison**. Tables 1 show that our CS and LS methods can consistently enhance the widely used UNet and UViT backbones under the unconditional [3–5], class-conditional [36, 37] and text-to-image [38–42] settings with almost no additional computational cost. Moreover, on both UNet and UViT, our CS and LS also achieve much better performance than $1/\sqrt{2}$-CS. All these results are consistent with the analysis in Section 4.1.

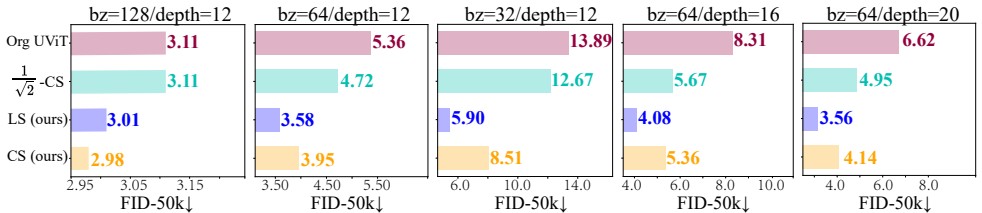

Figure 6: Synthesis performance of proposed methods under small batch size and deep architecture.

**Robustness to Batch Size and Network Depth**. UViT is often sensitive to batch size and network depth [36], since small batch sizes yield more noisy stochastic gradients that further aggravate gradient instability. Additionally, Section 3 shows that increasing network depth impairs the stability of both forward and backward propagation and its robustness to noisy inputs. So in Fig. 6, we evaluate our methods on standard UViT with different batch sizes and network depths. The results show that our CS and LS methods can greatly help UViT on CIFAR10 by showing big improvements on standard UViT and $1/\sqrt{2}$-CS-based UViT under different training batch sizes and network depth.

**Robustness of LS to Network Design**. For the calibration module in LS, here we investigate the robustness of LS to the module designs. Table 2 shows that using advanced modules, e.g. IE [77, 80] and SE [79], can

Table 2: Robustness of LS to network design

| Ratio $r$ | FID↓ | Activation | FID↓ | Modules | FID↓ |
|---|---|---|---|---|---|
| 4 | 3.06 | ELU [75] | 3.08 | Learnable $\kappa_i$ | 3.46 |
| 8 | 3.10 | ReLU [76] (ours) | **3.01** | IE [77] | 3.09 |
| 16 (ours) | **3.01** | Mish [78] | 3.08 | SE [79] (ours) | **3.01** |

improve LS performance compared with simply treating $\kappa_i$ as learnable coefficients. So we use SE module [79, 81, 82] in our experiments (see Appendix). Furthermore, we observe that LS is robust to different activation functions and compression ratios $r$ in the SE module. Hence, our LS does not require elaborated model crafting and adopts default $r = 16$ and ReLU throughout this work.

**Other scaling methods**. Our LS and CS methods scale LSC, unlike previous scaling methods [46, 48–53] that primarily scale block output in classification tasks. Table 3 reveals that these block-output scaling methods have worse performance than LS (FID: 3.01) and CS (FID: 2.98) under diffusion model setting with UViT as backbone.

Table 3: Other scaling

| Mehtod (CIFAR10) | FID↓ |
|---|---|
| stable $\tau = 0.1$ [46] | 3.77 |
| stable $\tau = 0.3$ [46] | 3.75 |
| ReZero [48] | 3.46 |

## 6 Discussion

**Relationship between LS and CS**. Taking UViT on CIFAR10, MS-COCO, ImageNet and CelebA as baselines, we randomly sample 30 Gaussian noises from $\mathcal{N}(0, \mathbf{I})$ and measure the scaling coefficients, i.e., $\kappa_i \leq 1$, learned by LS for each LSC. Fig. 7 shows that LS and CS share similar characteristics. Firstly, the predicted coefficients of LS fall within the orange area which is the estimated coefficient range given by CS in Section 4.1. Moreover, these learned coefficients $\kappa_i$ share the almost same exponential curve with Eq. (8) in CS. These shared characteristics have the potential to serve as crucial starting points for further optimization of DM network architectures in the future. Moreover, we try to preliminary analyze these shared characteristics.

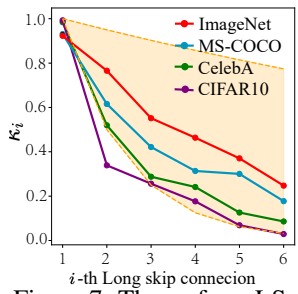

Figure 7: The $\kappa_i$ from LS.

First, for CS, how is the direction of applying the exponentially decayed scale determined? In fact, if we use the reverse direction, namely, $\kappa_i = \kappa^{N-i+1}$ ($\kappa < 1$), the stability bound in Theorem 3.4 is extremely large. The main term of stability bound in Theorem 3.4 can be written as $\mathcal{S}_r = \sum_{i=1}^{N} \kappa_i M_0^i = \kappa^N M_0 + \kappa^{N-1} M_0^2 + ... + \kappa M_0^N$, and could be very large, since $M_0^N > 1$ is large when $N$ is large and scaling it by a factor $\kappa$ could not sufficiently control its magnitude. In contrast, our default setting $\kappa_i = \kappa^{i-1}$ of CS can well control the main term in stability bound: $\mathcal{S} = \sum_{i=1}^{N} \kappa_i M_0^i = M_0 + \kappa^1 M_0^2 + ... + \kappa^{N-1} M_0^N$, where the larger terms $M_0^{i+1}$ are weighted by smaller coefficients $\kappa^i$. In this way, $\mathcal{S}$ is much smaller than $\mathcal{S}_r$, which shows the advantages of our default setting. Besides, the following Table 4 also compares the above two settings by using UViT on Cifar10 (batch size = 64), and shows that our default setting exhibits significant advantages.

Next, for LS, there are two possible reasons for why LS discovers a decaying scaling curve similar to the CS. On the one hand, from a theoretical view, as discussed for the direction of scaling, for the $i$-th long skip connection ($1 \leq i \leq N$), the learnable $\kappa_i$ should be smaller to better control the

Table 4: The FID-10k↓ result of different direction of scaling.

| Training step | 5k | 10k | 15k | 20k | 25k | 30k | 35k | 40k | 45k |
|---|---|---|---|---|---|---|---|---|---|
| $\kappa_i = \kappa^{N-i+1}$ | 67.26 | 33.93 | 22.78 | 16.91 | 15.01 | 14.01 | 12.96 | 12.34 | 12.26 |
| $\kappa_i = \kappa^{i-1}$ (ours) | 85.19 | 23.74 | 15.36 | 11.38 | 10.02 | 8.61 | 7.92 | 7.27 | 6.65 |

magnitude of $M_0^i$ so that the stability bound, e.g. in Theorem 3.4, is small. This directly yields the decaying scaling strategy which is also learnt by the scaling network. On the other hand, we can also analyze this observation in a more intuitive manner. Specifically, considering the UNet architecture, the gradient that travels through the $i$-th long skip connection during the backpropagation process influences the updates of both the first $i$ blocks in the encoder and the last $i$ blocks in the UNet decoder. As a result, to ensure stable network training, it's advisable to slightly reduce the gradients on the long skip connections involving more blocks (i.e., those with larger $i$ values) to prevent any potential issues with gradient explosion.

**Limitations**. CS does not require any additional parameters or extra computational cost. But it can only estimate a rough range of the scaling coefficient $\kappa$ as shown in Section 4.1. This means that one still needs to manually select $\kappa$ from the range. In the future, we may be able to design better methods to assist DMs in selecting better values of $\kappa$, or to have better estimates of the range of $\kappa$. Another effective solution is to use our LS method which can automatically learn qualified scaling coefficient $\kappa_i$ for each LSC. But LS inevitably introduces additional parameters and computational costs. Luckily, the prediction network in LS is very tiny and has only about 0.01M parameters, working very well in all our experiments.

Moreover, our CS and LS mainly focus on stabilizing the training of DMs, and indeed well achieve their target: greatly improving the stability of UNet and UViT whose effects is learning speed boosting of them by more than $1.5\times$ as shown in Fig. 5. But our CS and LS do not bring a very big improvement in terms of the final FID performance in Table 1, although they consistently improve model performance. Nonetheless, considering their simplicity, versatility, stabilizing ability to DMs, and faster learning speed, LS and CS should be greatly valuable in DMs.

**Conclusion**. We theoretically analyze the instability risks from widely used standard UNet for diffusion models (DMs). These risks are about the stability of forward and backward propagation, as well as the robustness of the network to extra noisy inputs. Based on these theoretical results, we propose a novel framework ScaleLong including two effective scaling methods, namely LS and CS, which can stabilize the training of DMs in different settings and lead to faster training speed and better generation performance.

**Acknowledgments**. This work was supported in part by National Key R&D Program of China under Grant No.2021ZD0111601, National Natural Science Foundation of China (NSFC) under Grant No.61836012, 62325605, U21A20470, GuangDong Basic and Applied Basic Research Foundation under Grant No. 2023A1515011374, GuangDong Province Key Laboratory of Information Security Technology.

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

The appendix is structured as follows. In Appendix A, we provide the training details and network architecture design for all the experiments conducted in this paper. Additionally, we present the specific core code used in our study. Appendix B presents the proofs for all the theorems and lemmas introduced in this paper. Finally, in Appendix C, we showcase additional examples of feature oscillation to support our argument about the training instability of existing UNet models when training diffusion models. Furthermore, we investigate the case of scaling coefficient $\kappa$ greater than 1 in CS, and the experimental results are consistent with our theoretical analysis.

# A Implementation details

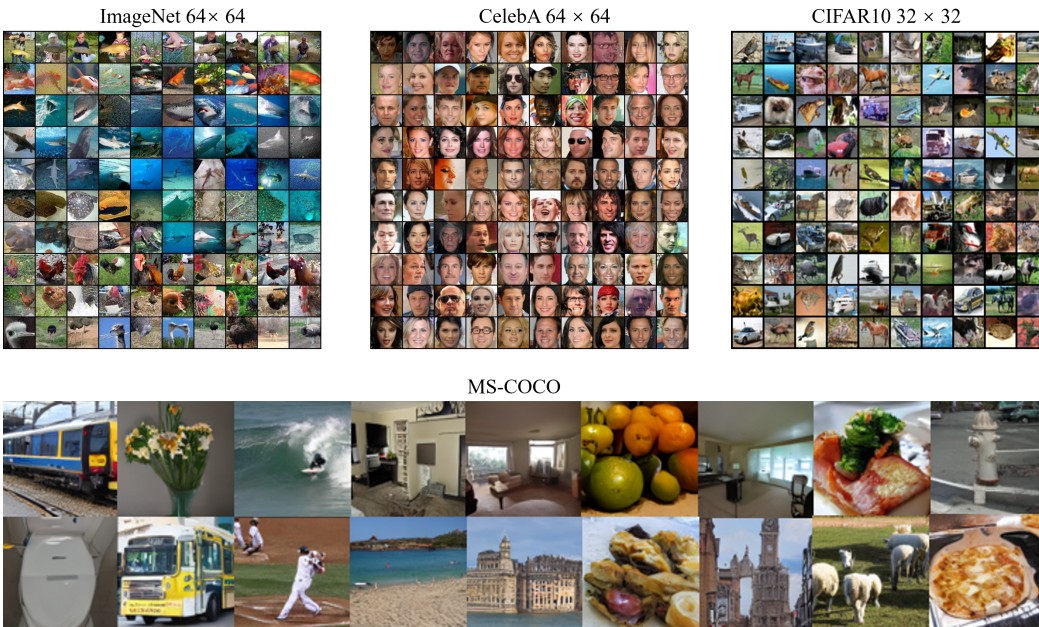

Figure 8: The examples of the generated images by UViT with our proposed LS.

## A.1 Experimental setup

In this paper, we consider four commonly used datasets for our experiments: CIFAR10 [32], CelebA [33], ImageNet [34], and MS-COCO [35]. For the unconditional setting, we use CIFAR10, which consists of 50,000 training images, and CelebA 64, which contains 162,770 training images of human faces. For the class-conditional settings, we utilize the well-known ImageNet dataset at 64 $\times$ 64 resolutions, which contains 1,281,167 training images from 1,000 different classes. For the text-to-image setting, we consider the MS-COCO dataset at 256 $\times$ 256 resolution. Unless otherwise specified, all of our experiments are conducted on A100 GPU and follow the experimental settings in [36].

Table 5: The summary of the datasets for the experiments.

| Dataset | #class | #training | #type | #Image size |
|---------|--------|-----------|-------|-------------|
| CIFAR10 | 10 | 50,000 | Uncondition | 32 x 32 |
| CelebA | 100 | 162,770 | Uncondition | 64 x 64 |
| ImageNet 64 | 1000 | 1,281,167 | Class-condition | 64 x 64 |
| MS-COCO | - | 82,783 | Text-to-Image | 256 x 256 |

## A.2 Network architecture details.

The scaling strategies we proposed on long skip connections, namely CS and LS, are straightforward to implement and can be easily transferred to neural network architectures of different diffusion models. Fig. 9 and Fig. 10 compare the original UViT with the code incorporating CS or LS, demonstrating that our proposed methods require minimal code modifications.

Table 6: The training settings for different dataset.

| Dataset | CIFAR10 64×64 | CelebA 64×64 | ImageNet 64×64 | MS-COCO |
|---|---|---|---|---|
| Latent shape | - | - | - | 32×32×4 |
| Batch size | 128/64/32 | 128 | 1024 | 256 |
| Training iterations | 500k | 500k | 300k | 1M |
| Warm-up steps | 2.5k | 4k | 5k | 5k |
| Optimizer | AdamW | AdamW | AdamW | AdamW |
| Learning rate | 2.00E-04 | 2.00E-04 | 3.00E-04 | 2.00E-04 |
| Weight decay | 0.03 | 0.03 | 0.03 | 0.03 |
| Beta | (0.99,0.999) | (0.99,0.99) | (0.99,0.99) | (0.9,0.9) |

Origin

```
skips = []
for blk in self.in_blocks:
    x = blk(x)
    skips.append(x)

x = self.mid_block(x)

for blk in self.out_blocks:
    x = blk(x, skips.pop())

x = self.norm(x)
x = self.decoder_pred(x)
assert x.size(1) == self.extras + L
x = x[:, self.extras:, :]
x = unpatchify(x, self.in_chans)
x = self.final_layer(x)
return x
```

CS (ours)

```
skips = []
for blk in self.in_blocks:
    x = blk(x)
    skips.append(x)

x = self.mid_block(x)

layer_cnt = 0
for blk in self.out_blocks:
    x = blk(x, skips.pop() * k** (self.depth - 1 - layer_cnt) )
    layer_cnt += 1

x = self.norm(x)
x = self.decoder_pred(x)
assert x.size(1) == self.extras + L
x = x[:, self.extras:, :]
x = unpatchify(x, self.in_chans)
x = self.final_layer(x)
return x
```

Figure 9: The comparison between the original UViT and our proposed CS.

CS requires no additional parameters and can rapidly adapt to any UNet-alike architecture and dataset. In our experiments, we have chosen values of $\kappa$ as 0.5, 0.5, 0.7, and 0.8 for the four datasets illustrated in Table 5, respectively. In various training environments, we recommend setting $\kappa$ within the range of 0.5 to 0.95. LS designs a calibration network shared by all long skip connections to predict scaling coefficients $\{\kappa_i\}$. Specifically, for the $i$-th long skip connection, its input feature is $x_i \in \mathbb{R}^{B \times N \times D}$, where $B$ denotes the batch size. For convolution-based UNet, $N$ and $D$ respectively denote the channel number and spatial dimension ($H \times W$) of the feature map; for transformer-based UViT [36], $N$ and $D$ respectively represent the token number and token dimension.

In the main text, we define the LS as

$$\kappa_i = \sigma(\zeta_\phi[\text{GAP}(x_i)]) \in \mathbb{R}^{B \times N \times 1}, 1 \leq i \leq N, \tag{10}$$

For $\zeta_\phi$, we employ a neural network structure based on SENet [79], given by:

$$\zeta_\phi(x) = \mathbf{W}_1 \circ \psi(\mathbf{W}_2 \circ x), \tag{11}$$

where $x \in \mathbb{R}^N$, $\circ$ denotes matrix multiplication, and $\psi$ represents the ReLU activation function. The matrices $\mathbf{W}_1 \in \mathbb{R}^{N \times N/r}$ and $\mathbf{W}1 \in \mathbb{R}^{N/r \times N}$, where $r$ is the compression ratio, are set to $r = 16$ in this paper. In the ablation study of the main text, we have demonstrated that the choice of $r$ has a minimal impact on the performance of the diffusion model. It is worth noting that if LS is applied to long skip connections where the feature dimensions remain unchanged, such as in UViT, then the calibration network is only a shared $\zeta_\phi(x)$.

However, when LS is applied to neural networks where the dimension of features in certain long skip connections may change, such as in UNet with downsample and upsample operations, we need to provide a set of encoders and decoders in the calibration network to ensure dimension alignment for different feature dimensions.

Assuming there are $d$ different feature dimensions in the long skip connections, where the dimension of the $i$-th feature is ($B \times N_i \times D$), $1 \leq i \leq d$, the input dimension of $\zeta_\phi(x)$ is set to $N_{\min} = \min N_i$. Additionally, we set up $d$ sets of encoders and decoders, where features with the same dimension

|  Origin | LS (ours) |
| --- | --- |

```python
skips = []
for blk in self.in_blocks:
    x = blk(x)
    skips.append(x)

x = self.mid_block(x)

for blk in self.out_blocks:
    x = blk(x, skips.pop())

x = self.norm(x)
x = self.decoder_pred(x)
assert x.size(1) == self.extras + L
x = x[:, self.extras:, :]
x = unpatchify(x, self.in_chans)
x = self.final_layer(x)
return x
```

```python
skips = []
for blk in self.in_blocks:
    x = blk(x)
    skips.append(x)

x = self.mid_block(x)

for blk in self.out_blocks:
    skip = skips.pop()
    x = blk(x, skip * self.LS(skip) )

x = self.norm(x)
x = self.decoder_pred(x)
assert x.size(1) == self.extras + L
x = x[:, self.extras:, :]
x = unpatchify(x, self.in_chans)
x = self.final_layer(x)
return x
```

Figure 10: The comparison between the original UViT and our proposed LS.

share an encoder and a decoder. The $i$-th set of encoder $\zeta_{\text{en}}^{(i)}$ and decoder $\zeta_{\text{de}}^{(i)}$ also adopt the SENet structure, given by:

$$\zeta_{\text{en}}^{(i)}(x) = \mathbf{W}_{N_{\min} \times N_i/r_i} \circ \psi(\mathbf{W}_{N_i/r_i \times N_i} \circ x), \quad x \in \mathbb{R}^{N_i} \tag{12}$$

where $r_i$ is the compression ratio, typically set to $\lfloor N_i/4 \rfloor$, and

$$\zeta_{\text{de}}^{(i)}(x) = \mathbf{W}_{N_i \times N_i/r_i} \circ \psi(\mathbf{W}_{N_i/r_i \times N_{\min}} \circ x), \quad x \in \mathbb{R}^{N_{\min}}. \tag{13}$$

In fact, since the parameter count of $\zeta_\phi(x)$ itself is not substantial, to make LS more versatile, we can also individually set a scaling module into each long skip connection. This configuration typically does not lead to significant performance degradation and can even result in improved performance, while maintaining the same inference speed as the original LS.

# B  The proof of the Theorems and Lemma

## B.1  Preliminaries

**Diffusion Model (DM)**. DDPM-alike DMs [1–6, 8, 9, 43] generates a sequence of noisy samples $\{\mathbf{x}_i\}_{i=1}^T$ by repeatedly adding Gaussian noise to a sample $\mathbf{x}_0$ until attatining $\mathbf{x}_T \sim \mathcal{N}(\mathbf{0}, \mathbf{I})$. This noise injection process, a.k.a. the forward process, can be formalized as a Markov chain $q(\mathbf{x}_{1:T}|\mathbf{x}_0) = \prod_{t=1}^T q(\mathbf{x}_t|\mathbf{x}_{t-1})$, where $q(\mathbf{x}_t|\mathbf{x}_{t-1}) = \mathcal{N}(\mathbf{x}_t|\sqrt{\alpha_t}\mathbf{x}_{t-1}, \beta_t\mathbf{I})$, $\alpha_t$ and $\beta_t$ depend on $t$ and satisfy $\alpha_t + \beta_t = 1$. By using the properties of Gaussian distribution, the forward process can be written as

$$q(\mathbf{x}_t|\mathbf{x}_0) = \mathcal{N}(\mathbf{x}_t; \sqrt{\bar{\alpha}_t}\mathbf{x}_0, (1 - \bar{\alpha}_t)\mathbf{I}), \tag{14}$$

where $\bar{\alpha}_t = \prod_{i=1}^t \alpha_i$. Next, one can sample $\mathbf{x}_t = \sqrt{\bar{\alpha}_t}\mathbf{x}_0 + \sqrt{1 - \bar{\alpha}_t}\epsilon_t$, where $\epsilon_t \sim \mathcal{N}(\mathbf{0}, \mathbf{I})$. Then DM adopts a neural network $\hat{\epsilon}_\theta(\cdot, t)$ to invert the forward process and predict the noise $\epsilon_t$ added at each time step. This process is to recover data $\mathbf{x}_0$ from a Gaussian noise by minimizing the loss

$$\ell_{\text{simple}}^t(\theta) = \mathbb{E}\|\epsilon_t - \hat{\epsilon}_\theta(\sqrt{\bar{\alpha}_t}\mathbf{x}_0 + \sqrt{1 - \bar{\alpha}_t}\epsilon_t, t)\|_2^2. \tag{15}$$

**UNet-alike Network**. Since the network $\hat{\epsilon}_\theta(\cdot, t)$ in DMs predicts the noise to denoise, it plays a similar role of UNet-alike network widely in image restoration tasks, e.g. image de-raining, image denoising [22–26, 44, 45]. This inspires DMs to use UNet-alike network in (16) that uses LSCs to connect distant parts of the network for long-range information transmission and aggregation

$$\mathbf{UNet}(x) = f_0(x), \quad f_i(x) = b_{i+1} \circ [\kappa_{i+1} \cdot a_{i+1} \circ x + f_{i+1}(a_{i+1} \circ x)], \ 0 \le i \le N - 1 \tag{16}$$

where $x \in \mathbb{R}^m$ denotes the input, $a_i$ and $b_i$ ($i \ge 1$) are the trainable parameter of the $i$-th block, $\kappa_i > 0$ ($i \ge 1$) are the scaling coefficients and are set to 1 in standard UNet. $f_N$ is the middle block of UNet. For the vector operation $\circ$, it can be designed to implement different networks.

W.l.o.g, in this paper, we consider $\circ$ as matrix multiplication, and set the block $a_i$ and $b_i$ as the stacked networks [46, 47] which is implemented as

$$a_i \circ x = \mathbf{W}_l^{a_i}\phi(\mathbf{W}_{l-1}^{a_i}...\phi(\mathbf{W}_1^{a_i}x)), \quad b_i \circ x = \mathbf{W}_l^{b_i}\phi(\mathbf{W}_{l-1}^{b_i}...\phi(\mathbf{W}_1^{b_i}x)) \tag{17}$$

with a ReLU activation function $\phi$ and learnable matrices $\mathbf{W}_j^{a_i}, \mathbf{W}_j^{b_i} \in \mathbb{R}^{m \times m}$ ($j \ge 1$). Moreover, let $f_N$ also have the same architecture, i.e. $f_N(x) = \mathbf{W}_l^{f_i}\phi(\mathbf{W}_{l-1}^{f_i}...\phi(\mathbf{W}_1^{f_i}x))$.

**Proposition B.1** (Amoroso distribution). *The Amoroso distribution is a four parameter, continuous, univariate, unimodal probability density, with semi-infinite range [83]. And its probability density function is*

$$\mathbf{Amoroso}(X|a, \theta, \alpha, \beta) = \frac{1}{\Gamma(\alpha)}|\frac{\beta}{\theta}|(\frac{X - a}{\theta})^{\alpha\beta-1}\exp\left\{-(\frac{X - a}{\theta})^\beta\right\}, \tag{18}$$

*for $x, a, \theta, \alpha, \beta \in \mathbb{R}, \alpha > 0$ and range $x \ge a$ if $\theta > 0$, $x \le a$ if $\theta < 0$. The mean and variance of Amoroso distribution are*

$$\mathbb{E}_{X \sim \mathbf{Amoroso}(X|a, \theta, \alpha, \beta)}X = a + \theta \cdot \frac{\Gamma(\alpha + \frac{1}{\beta})}{\Gamma(\alpha)}, \tag{19}$$

*and*

$$\mathbf{Var}_{X \sim \mathbf{Amoroso}(X|a, \theta, \alpha, \beta)}X = \theta^2\left[\frac{\Gamma(\alpha + \frac{2}{\beta})}{\Gamma(\alpha)} - \frac{\Gamma(\alpha + \frac{1}{\beta})^2}{\Gamma(\alpha)^2}\right]. \tag{20}$$

**Proposition B.2** (Stirling's formula). *For big enough $x$ and $x \in \mathbb{R}^+$, we have an approximation of Gamma function:*

$$\Gamma(x + 1) \approx \sqrt{2\pi x}\left(\frac{x}{e}\right)^x. \tag{21}$$

**Proposition B.3** (Scaled Chi distribution). *Let $X = (x_1, x_2, ...x_k)$ and $x_i, i = 1, ..., k$ are $k$ independent, normally distributed random variables with mean 0 and standard deviation $\sigma$. The statistic $\ell_2(X) = \sqrt{\sum_{i=1}^k x_i^2}$ follows Scaled Chi distribution [83]. Moreover, $\ell_2(X)$ also follows* **Amoroso**$(x|0, \sqrt{2}\sigma, \frac{k}{2}, 2)$. *By Eq. (19) and Eq. (20), the mean and variance of Scaled Chi distribution are*

$$\mathbb{E}_{X \sim N(\mathbf{0}, \sigma^2 \cdot \mathbf{I_k})}[\ell_2(X)]^j = 2^{j/2}\sigma^j \cdot \frac{\Gamma(\frac{k+j}{2})}{\Gamma(\frac{k}{2})}, \tag{22}$$

*and*

$$\mathbf{Var}_{X \sim N(\mathbf{0}, \sigma^2 \cdot \mathbf{I_k})}\ell_2(X) = 2\sigma^2 \left[ \frac{\Gamma(\frac{k}{2}+1)}{\Gamma(\frac{k}{2})} - \frac{\Gamma(\frac{k+1}{2})^2}{\Gamma(\frac{k}{2})^2} \right]. \tag{23}$$

**Lemma B.4.** *For big enough $x$ and $x \in \mathbb{R}^+$, we have*

$$\lim_{x \to +\infty} \left[ \frac{\Gamma(\frac{x+1}{2})}{\Gamma(\frac{x}{2})} \right]^2 \cdot \frac{1}{x} = \frac{1}{2}. \tag{24}$$

*And*

$$\lim_{x \to +\infty} \frac{\Gamma(\frac{x}{2}+1)}{\Gamma(\frac{x}{2})} - \left[ \frac{\Gamma(\frac{x+1}{2})}{\Gamma(\frac{x}{2})} \right]^2 = \frac{1}{4}. \tag{25}$$

*Proof.*

$$\lim_{x \to +\infty} \left[ \frac{\Gamma(\frac{x+1}{2})}{\Gamma(\frac{x}{2})} \right]^2 \cdot \frac{1}{x} \approx \lim_{x \to +\infty} \left( \frac{\sqrt{2\pi(\frac{x-1}{2})} \cdot (\frac{x-1}{2e})^{\frac{x-1}{2}}}{\sqrt{2\pi(\frac{x-2}{2})} \cdot (\frac{x-2}{2e})^{\frac{x-2}{2}}} \right)^2 \cdot \frac{1}{x} \qquad \text{from Proposition. B.2}$$

$$= \lim_{x \to +\infty} \left( \frac{x-1}{x-2} \right) \cdot \frac{(\frac{x-1}{2e})^{x-2}}{(\frac{x-2}{2e})^{x-2}} \cdot \left( \frac{x-1}{2e} \right) \cdot \frac{1}{x}$$

$$= \lim_{x \to +\infty} \left( 1 + \frac{1}{x-2} \right)^{x-2} \cdot \frac{x-1}{x-2} \cdot \frac{x-1}{2e} \cdot \frac{1}{x}$$

$$= \frac{1}{2}$$

on the other hand, we have

$$\lim_{x \to +\infty} \frac{\Gamma(\frac{x}{2}+1)}{\Gamma(\frac{x}{2})} - \left[ \frac{\Gamma(\frac{x+1}{2})}{\Gamma(\frac{x}{2})} \right]^2 = \lim_{x \to +\infty} \frac{x}{2} - \left( 1 + \frac{1}{x-2} \right)^{x-2} \cdot \frac{x-1}{x-2} \cdot \frac{x-1}{2e}$$

$$= \lim_{x \to +\infty} \frac{x}{2e} \left( e - (1 + \frac{1}{x})^x \right)$$

$$= \frac{1}{2} \left( -\frac{\frac{1}{e}(-e)}{2} \right)$$

$$= \frac{1}{4}$$

□

**Lemma B.5.** *[84, 85] Let $\mathbf{x} = (x_1, x_2, ..., x_N)$, where $x_i, 1 \le i \le N$, are independently and identically distributed random variables with $x_i \sim \mathcal{N}(\mu, \sigma^2)$. In this case, the mathematical expectation $\mathbb{E}(\sum_{i=1}^N x_i^2)$ can be calculated as $\sigma^2 N + \mu^2 N$.*

**Lemma B.6.** *[86] Let $F_{ij} \in \mathbb{R}^{N_i \times k \times k}$ be the $j$-th filter of the $i$-th convolutional layer. If all filters are initialized as Kaiming's initialization, i.e., $\mathcal{N}(0, \frac{2}{N_i \times k \times k})$, in $i$-th layer, $F_{ij}$ ($j = 1, 2, ..., N_{i+1}$) are i.i.d and approximately follow such a distribution during training:*

$$F_{ij} \sim \mathcal{N}(\mathbf{0}, \mathcal{O}[(k^2 N_i)^{-1}] \cdot \mathbf{I}_{N_i \times k \times k}). \tag{26}$$

**Lemma B.7.** *[47, 87] For the feedforward neural network $\mathbf{W}_l \phi(\mathbf{W}_{l-1}...\phi(\mathbf{W}_1 x))$ with initialization $\mathcal{N}(0, \frac{2}{m})$, and $W_i, 1 \leq i \leq l, \in \mathbb{R}^{m \times m}$. $\forall \rho \in (0, 1)$, the following inequality holds with at least $1 - \mathcal{O}(l) \exp[-\Omega(m\rho^2/l)]$:*

$$(1 - \rho)\|h_0\|_2 \leq \|h_a\|_2 \leq (1 + \rho)\|h_0\|_2, \quad a \in \{1, 2, ..., l-1\}, \tag{27}$$

*where $h_a$ is $\phi(\mathbf{W}_a h_{a-1})$ for $a = 1, 2, ..., l-1$ with $h_0 = x$.*

**Lemma B.8.** *For a matrix $\mathbf{W} \in \mathbb{R}^{m \times m}$ and its elements are independently follow the distribution $\mathcal{N}(0, c^2)$, we have the follwing estimation while $m$ is large enough*

$$\|\mathbf{W}s\|_2 \approx \mathcal{O}(\|s\|_2), \tag{28}$$

*where $s \in \mathbb{R}^m$.*

*Proof.* Let $v_i, 1 \leq i \leq m$ be the $i$-th column of matrix $\mathbf{W}$, and we have $v_i \sim \mathcal{N}(\mathbf{0}, c^2 \mathbf{I}_m)$. We first prove the following two facts while $m \to \infty$:

(1) $\|v_i\|_2 \approx \|v_j\|_2 \to \sqrt{2}c \cdot \frac{\Gamma((m+1)/2)}{\Gamma(m/2)}, 1 \leq i < j \leq m$;

(2) $\langle v_i, v_j \rangle \to 0, 1 \leq i < j \leq m$;

For the fact (1), since Chebyshev inequality, for $1 \leq i \leq m$ and a given $M$, we have

$$P\left\{ |\|v_i\|_2 - \mathbb{E}(\|v_i\|_2)| \geq \sqrt{M \mathbf{Var}(\|v_i\|_2)} \right\} \leq \frac{1}{M}. \tag{29}$$

from Proposition B.3 and Lemma. B.4, we can rewrite Eq. (29) when $m \to \infty$:

$$P\left\{ \|v_i\|_2 \in \left[ \sqrt{2}c \cdot \frac{\Gamma((m+1)/2)}{\Gamma(m/2)} - \sqrt{\frac{M}{2}}c, \sqrt{2}c \cdot \frac{\Gamma((m+1)/2)}{\Gamma(m/2)} + \sqrt{\frac{M}{2}}c \right] \right\} \geq 1 - \frac{1}{M}. \tag{30}$$

For a small enough $\epsilon > 0$, let $M = 1/\epsilon$. Note that $\sqrt{\frac{M}{2}}c = c/\sqrt{2\epsilon}$ is a constant. When $m \to \infty$, $\sqrt{2}c \cdot \frac{\Gamma((m+1)/2)}{\Gamma(m/2)} \gg \sqrt{\frac{M}{2}}c$. Hence, for any $i \in [1, m]$ and any small enough $\epsilon$, we have

$$P\left\{ \|v_i\|_2 \approx \sqrt{2}c \cdot \frac{\Gamma((m+1)/2)}{\Gamma(m/2)} \right\} \geq 1 - \epsilon. \tag{31}$$

So the fact (1) holds. Moreover, we consider the $\langle v_i, v_j \rangle, 1 \leq i < j \leq m$.

Let $v_i = (v_{i1}, v_{i2}, ..., v_{im})$ and $v_j = (v_{j1}, v_{j2}, ..., v_{jm})$. So $\langle v_i, v_j \rangle = \sum_{p=1}^m v_{ip}v_{jp}$. Note that, $v_i$ and $v_j$ are independent, hence

$$\mathbb{E}(v_{ip}v_{jp}) = 0, \tag{32}$$

$$\mathbf{Var}(v_{ip}v_{jp}) = \mathbf{Var}(v_{ip})\mathbf{Var}(v_{jp}) + (\mathbb{E}(v_{ip}))^2 \mathbf{Var}(v_{jp}) + (\mathbb{E}(v_{jp}))^2 \mathbf{Var}(v_{ip}) = c^4, \tag{33}$$

since central limit theorem, we have

$$\sqrt{m} \cdot c^{-2} \cdot \left( \frac{1}{m} \sum_{p=1}^m v_{ip}v_{jp} - 0 \right) \sim N(0, 1), \tag{34}$$

According to Eq. (22), Lemma B.4 and Eq. (34), when $m \to \infty$, we have

$$\frac{\langle v_i, v_j \rangle}{\|v_i\|_2 \cdot \|v_j\|_2} \to \frac{1}{\sqrt{m}} \cdot \frac{\langle v_i, v_j \rangle}{\sqrt{m}} \sim N(0, \mathcal{O}(\frac{1}{m})) \to N(0, 0). \tag{35}$$

Therefore, the fact (2) holds and we have $\langle v_i, v_j \rangle \to 0, 1 \le i < j \le m$.

Next, we consider $\|\mathbf{W}s\|_2$ for any $s \in \mathbb{R}^m$.

$$
\begin{aligned}
\|\mathbf{W}s\|_2^2 &= s^{\mathrm{T}}[v_1, v_2, ..., v_m]^{\mathrm{T}}[v_1, v_2, ..., v_m]s \\
&= s^{\mathrm{T}}
\begin{pmatrix}
\|v_1\|_2^2 & v_1^{\mathrm{T}}v_2 & \cdots & v_1^{\mathrm{T}}v_m \\
v_2^{\mathrm{T}}v_1 & \|v_2\|_2^2 & & \vdots \\
\vdots & & \ddots & \\
v_m^{\mathrm{T}}v_1 & \cdots & & \|v_m\|_2^2
\end{pmatrix}
s
\end{aligned}
\tag{36}
$$

From the facts (1) and (2), when $m$ is large enough, we have $v_i^{\mathrm{T}}v_j \to 0, i \ne j$, and $\|v_1\|_2^2 \approx \|v_2\|_2^2 \approx \cdots \approx \|v_m\|_2^2 := c_v$. Therefore, we have

$$
\|\mathbf{W}s\|_2^2 = s^{\mathrm{T}}c_v\mathbf{I_m}s = \mathcal{O}(\|s\|_2^2).
\tag{37}
$$

$\square$

Let $k = 1$ in Lemma B.6, and the convolution kernel degenerates into a learnable matrix. In this case, we can observe that if $W \in \mathbb{R}^{m \times m}$ is initialized using Kaiming initialization, i.e., $\mathcal{N}(0, 2/m)$, during the training process, the elements of $W$ will approximately follow a Gaussian distribution $\mathcal{N}(0, \mathcal{O}(m^{-1}))$. In this case, we can easily rewrite Lemma B.7 as the following Corollary B.9:

**Corollary B.9.** *For the feedforward neural network $\mathbf{W}_l\phi(\mathbf{W}_{l-1}...\phi(\mathbf{W}_1 x))$ with kaiming's initialization during training. $\forall \rho \in (0, 1)$, the following inequality holds with at least $1 - \mathcal{O}(l)\exp[-\Omega(m\rho^2/l)]$:*

$$
\mathcal{O}((1-\rho)\|h_0\|_2) \le \|h_a\|_2 \le \mathcal{O}((1+\rho)\|h_0\|_2), \quad a \in \{1, 2, ..., l-1\},
\tag{38}
$$

*where $h_a$ is $\phi(\mathbf{W}_a h_{a-1})$ for $a = 1, 2, ..., l-1$ with $h_0 = x$.*

Moreover, considering Lemma B.6 and Lemma B.8, we can conclude that the matrix $W_l$ in Corollary B.9 will not change the magnitude of its corresponding input $\phi(\mathbf{W}_{l-1}...\phi(\mathbf{W}_1 x))$. In other words, we have the following relationship:

$$
\mathcal{O}((1-\rho)\|h_0\|_2) \le \|\mathbf{W}_l h_{l-1}\|_2 \le \mathcal{O}((1+\rho)\|h_0\|_2),
\tag{39}
$$

Therefore, for $a_i$ and $b_i$ defined in Eq. (17), we have following corollary:

**Corollary B.10.** *For $a_i$ and $b_i$ defined in Eq. (17), $1 \le i \le N$, and their corresponding input $x_a$ and $x_b$. $\forall \rho \in (0, 1)$, the following each inequality holds with at least $1 - \mathcal{O}(l)\exp[-\Omega(m\rho^2/l)]$:*

$$
\|a_i \circ x_a\|_2 = \mathcal{O}(\|x_a\|_2), \quad \|b_i \circ x_b\|_2 = \mathcal{O}(\|x_b\|_2),
\tag{40}
$$

**Lemma B.11.** *[47] Let $\mathbf{D}$ be a diagonal matrix representing the ReLU activation layer. Assume the training loss is $\frac{1}{n}\sum_{s=1}^n \ell_s(\mathbf{W})$, where $\ell_s(\mathbf{W})$ denotes the training loss of the $s$-th sample among the $n$ training samples. For the $\mathbf{W}_p^q$ defined in Eq. (17) and $f_0(\mathbf{x}_t)$ defined in Eq. (16), where $p \in \{1, 2, ..., l\}, q \in \{a_i, b_i\}$, the following inequalities hold with at least $1 - \mathcal{O}(nN)\exp[-\Omega(m)]$:*

$$
\|\mathbf{D}(\mathbf{W}_{p+1}^{a_i})^{\mathrm{T}} \cdots \mathbf{D}(\mathbf{W}_l^{a_i})^{\mathrm{T}} \cdots \mathbf{D}(\mathbf{W}_l^{b_1})^{\mathrm{T}}v\|_2 \le \mathcal{O}(\|v\|_2),
\tag{41}
$$

*and*

$$
\|\mathbf{D}(\mathbf{W}_{p+1}^{b_i})^{\mathrm{T}} \cdots \mathbf{D}(\mathbf{W}_l^{b_1})^{\mathrm{T}}v\|_2 \le \mathcal{O}(\|v\|_2),
\tag{42}
$$

*where $v \in \mathbb{R}^m$.*

## B.2    The proof of Theorem 3.1.

**Theorem 3.1**. Assume that all learnable matrices of UNet in Eq. (16) are independently initialized as Kaiming's initialization, i.e., $\mathcal{N}(0, \frac{2}{m})$. Then for any $\rho \in (0, 1]$, by minimizing the training loss Eq. (15) of DMs, with probability at least $1 - \mathcal{O}(N) \exp[-\Omega(m\rho^2)]$, we have

$$(1 - \rho)^2 \left[ c_1 \|\mathbf{x}_t\|_2^2 \cdot \sum_{j=i+1}^{N} \kappa_j^2 + c_2 \right] \lesssim \|h_i\|_2^2 \lesssim (1 + \rho)^2 \left[ c_1 \|\mathbf{x}_t\|_2^2 \cdot \sum_{j=i+1}^{N} \kappa_j^2 + c_2 \right], \quad (43)$$

where the hidden feature $h_i$ is the output of $f_i$ defined in Eq. (16), $\mathbf{x}_t = \sqrt{\bar{\alpha}_t} \mathbf{x}_0 + \sqrt{1 - \bar{\alpha}_t} \epsilon_t$ is the input of UNet; $c_1$ and $c_2$ are two constants; $\kappa_i$ is the scaling coefficient of the $i$-th LSC in UNet.

*Proof.* Let $s$ represent the input of $f_i$ in Eq. (16), i.e., $s = a_i \circ a_{i-1} \circ \cdots a_1 \circ \mathbf{x}_t$, and let $h_i = f_i(s)$. By expanding Eq. (16), we can observe that $h_i = f_i(s)$ is mainly composed of the following $N - i$ terms in the long skip connection:

$$\begin{cases} \kappa_{i+1} \cdot b_{i+1} \circ a_{i+1} \circ s; \\ \kappa_{i+2} \cdot b_{i+1} \circ b_{i+2} \circ a_{i+2} \circ a_{i+1} \circ s; \\ \cdots \\ \kappa_N \cdot b_{i+1} \circ b_{i+2} \cdots \circ b_N \circ a_N \cdots a_{i+1} \circ s; \end{cases} \quad (44)$$

When $N$ is sufficiently large, $h_i$ is predominantly dominated by $N$ long skip connection terms, while the information on the residual term is relatively minor. Hence, we can approximate Eq. (16) as follows:

$$h_i = f_i(s) \approx \sum_{j=i+1}^{N} \kappa_j \cdot \pi_b^j \circ \pi_a^j \circ s, \quad 0 \le i \le N - 1, \quad (45)$$

where $\pi_b^j := b_{i+1} \circ b_{i+2} \cdots \circ b_j$, $\pi_a^j := a_j \circ a_{j-1} \cdots \circ a_{i+1}$.

In practice, the depth $l$ of block $a_i$ or $b_i$ is a universal constant, therefore, from Corollary B.9, $\forall \rho \in (0, 1)$ and $j \in \{i+1, i+2, ..., N\}$, with probability at least $1 - \mathcal{O}(N) \exp[-\Omega(m\rho^2)]$, we have

$$\|h_{\kappa_j}\|_2 \in \mathcal{O}((1 \pm \rho)\|\mathbf{x}_t\|_2), \quad (46)$$

where

$$h_{\kappa_j} = \phi(\mathbf{W}_{l-1}^{b_{i+1}}...\phi(\mathbf{W}_1^{b_{i+1}} r)), \quad r = b_{i+2} \cdots \circ b_j \circ \pi_a^j \circ s, \quad (47)$$

s.t. $h_i = f_i(s) \approx \sum_{j=i+1}^{N} \kappa_j \mathbf{W}_l^{b_{i+1}} \circ h_{\kappa_j}$. From Lemma B.6, the elements of $\mathbf{W}_l^{b_{i+1}}$ will approximately follow a Gaussian distribution $\mathcal{N}(0, \mathcal{O}(m^{-1}))$, therefore, $\mathbb{E}(\mathbf{W}_l^{b_{i+1}} \circ h_{\kappa_j}) = 0$. Moreover, from Lemma B.8,

$$\mathbf{Var}(\mathbf{W}_l^{b_{i+1}} \circ h_{\kappa_j}) \approx h_{\kappa_j}^T \mathcal{O}(m^{-1}) \cdot \mathbf{I}_{m \times m} h_{\kappa_j} = \mathcal{O}(m^{-1}) \|h_{\kappa_j}\|_2^2 \cdot \mathbf{I}_{m \times m}. \quad (48)$$

Therefore [47], $h_i \approx \sum_{j=i+1}^{N} \kappa_j \mathbf{W}_l^{b_{i+1}} \circ h_{\kappa_j}$ can be approximately regarded as a Gaussian variable with zero mean and covariance matrix $\sum_{j=i+1}^{N} \kappa_j^2 \mathcal{O}(m^{-1}) \|h_{\kappa_j}\|_2^2 \cdot \mathbf{I}_{m \times m}$. According to Lemma B.5, we have

$$\|h_i\|_2^2 \approx \mathbb{E}(\sum_{j=i+1}^{N} \kappa_j \mathbf{W}_l^{b_{i+1}} \circ h_{\kappa_j}) = m \cdot \sum_{j=i+1}^{N} \kappa_j^2 \mathcal{O}(m^{-1}) \|h_{\kappa_j}\|_2^2 = \sum_{j=i+1}^{N} \kappa_j^2 \mathcal{O}(1) \|h_{\kappa_j}\|_2^2. \quad (49)$$

From Eq. (46) and Eq. (49), we have

$$(1 - \rho)^2 \left[ c_1 \|\mathbf{x}_t\|_2^2 \cdot \sum_{j=i+1}^{N} \kappa_j^2 + c_2 \right] \lesssim \|h_i\|_2^2 \lesssim (1 + \rho)^2 \left[ c_1 \|\mathbf{x}_t\|_2^2 \cdot \sum_{j=i+1}^{N} \kappa_j^2 + c_2 \right], \quad (50)$$

where we use $c_1$ to denote $\mathcal{O}(1)$ in Eq. (49) and we use the constant $c_2$ to compensate for the estimation loss caused by all the approximation calculations in the derivation process.

$\square$

## B.3 The proof of Lemma 3.2.

---

**Lemma 3.2.** For $\mathbf{x}_t = \sqrt{\bar{\alpha}_t}\mathbf{x}_0 + \sqrt{1-\bar{\alpha}_t}\epsilon_t$ defined in Eq. (14) as a input of UNet, $\epsilon_t \sim \mathcal{N}(0,\mathbf{I})$, if $\mathbf{x}_0$ follow the uniform distribution $U[-1,1]$, then we have

$$\mathbb{E}\|\mathbf{x}_t\|_2^2 = (1 - 2\mathbb{E}_t\bar{\alpha}_t/3)m = \mathcal{O}(m). \tag{51}$$

---

*Proof.* For $\mathbf{x}_t = \sqrt{\bar{\alpha}_t}\mathbf{x}_0 + \sqrt{1-\bar{\alpha}_t}\epsilon_t$, we have

$$\mathbb{E}\|\mathbf{x}_t\|_2^2 = \mathbb{E}_{t,\mathbf{x}_0}(\bar{\alpha}_t \sum_{j=1}^m \mathbf{x}_0^2(j) + (1-\bar{\alpha}_t)m), \tag{52}$$

where $\mathbf{x}_0(j)$ represents the $j$-th element of $\mathbf{x}_0$. The equality in Eq. (52) follows from the fact that, from Eq. (14) and Lemma B.5, we know that $\mathbf{x}_t$ follows $\mathcal{N}(\mathbf{x}_t; \sqrt{\bar{\alpha}_t}\mathbf{x}_0, (1-\bar{\alpha}_t)\mathbf{I})$. Therefore, $\|\mathbf{x}_t\|_2^2$ follows a chi-square distribution, whose mathematical expectation is $\mathbb{E}_{t,\mathbf{x}_0}(\bar{\alpha}_t \sum_{j=1}^m \mathbf{x}_0^2(j) + (1 - \bar{\alpha}_t)m)$.

Moreover, since $\mathbf{x}_0$ follow the uniform distribution $U[-1,1]$, we have

$$\begin{aligned}
\mathbb{E}\|\mathbf{x}_t\|_2^2 &= \mathbb{E}_t\bar{\alpha}_t \cdot \sum_{j=1}^m \int_{-\infty}^{\infty} x^2 P_{x\sim U[-1,1]}(x)dx + \mathbb{E}_t(1-\bar{\alpha}_t)m \\
&= \mathbb{E}_t\bar{\alpha}_t \cdot m/3 + \mathbb{E}_t(1-\bar{\alpha}_t)m = (1 - 2\mathbb{E}_t\bar{\alpha}_t/3)m = \mathcal{O}(m).
\end{aligned} \tag{53}$$

where $P_{x\sim U[-1,1]}(x)$ is density function of the elements in $\mathbf{x}_0$. The last equality in Eq. (53) follows the fact that, according to the setup of the diffusion model, it is generally observed that $\bar{\alpha}_t$ monotonically decreases to 0 as $t$ increases, with $\bar{\alpha}_t \leq 1$. Consequently, we can easily deduce that $\mathbb{E}_t\bar{\alpha}_t$ is of the order $\mathcal{O}(1)$. For instance, in the setting of DDPM, we have $\mathbb{E}_t\bar{\alpha}_t = \mathbb{E}_t\left(\prod_{i=1}^t \sqrt{1 - \frac{0.02i}{1000}}\right) \approx 0.27 = \mathcal{O}(1)$.

$\square$

---

**Theorem 3.3.** Assume that all learnable matrices of UNet in Eq. (16) are independently initialized as Kaiming's initialization, i.e., $\mathcal{N}(0, \frac{2}{m})$. Then for any $\rho \in (0,1]$, with probability at least $1 - \mathcal{O}(nN)\exp[-\Omega(m)]$, for a sample $\mathbf{x}_t$ in training set, we have

$$\|\nabla_{\mathbf{W}_p^q}\ell_s(\mathbf{W})\|_2^2 \lesssim \mathcal{O}\left(\ell_s(\mathbf{W}) \cdot \|\mathbf{x}_t\|_2^2 \cdot \sum_{j=i}^N \kappa_j^2 + c_3\right), \quad (p \in \{1,2,...,l\}, q \in \{a_i, b_i\}),$$
(54)

where $\mathbf{x}_t = \sqrt{\bar{\alpha}_t}\mathbf{x}_0 + \sqrt{1-\bar{\alpha}_t}\epsilon_t$ denotes the noisy sample of the $s$-th sample, $\epsilon_t \sim \mathcal{N}(\mathbf{0},\mathbf{I})$, $N$ is the number of LSCs, $c_3$ is a small constant. $n$ is the size of the training set.

---

*Proof.* From Eq. (16) and Eq. (44), we can approximate the output of UNet $f_0(\mathbf{x}_t)$ by weight sum of $N$ forward path $g^j$, $1 \leq j \leq N$, i.e.,

$$f_0(\mathbf{x}_t) \approx \sum_{j=1}^N \kappa_i \cdot g^j. \tag{55}$$

where $g^j$ denote $b_1 \circ b_2 \circ \cdots b_j \circ a_j \circ \cdots a_1 \circ \mathbf{x}_t$, and $1 \leq j \leq N$.

Moreover, we use $g_p^{j,q}$ represent the feature map after $\mathbf{W}_p^q$ in forward path $g^j$, i.e., $g_p^{j,q} = \mathbf{W}_p^q h_{p-1}^{j,q}$ and $h_p^{j,q} = \phi(g_p^{j,q})$. Next, we first estimate $\|\nabla_{\mathbf{W}_p^{a_i}}\ell_s(\mathbf{W})\|_2^2$,

$$\|\nabla_{\mathbf{W}_p^{a_i}}\ell_s(\mathbf{W})\|_2^2 = \|\sum_{j=i}^{N} \partial\ell_s(\mathbf{W})/\partial g^j \cdot \partial g^j/\partial g_p^{j,a_i} \cdot \partial g_p^{j,a_i}/\partial\mathbf{W}_p^{a_i}\|_2^2$$

$$\leq \sum_{j=i}^{N} \|\partial\ell_s(\mathbf{W})/\partial g^j \cdot \partial g^j/\partial g_p^{j,a_i} \cdot \partial g_p^{j,a_i}/\partial\mathbf{W}_p^{a_i}\|_2^2$$

$$\leq \sum_{j=i}^{N} \|\mathbf{D}(\mathbf{W}_{p+1}^{a_i})^{\mathrm{T}}\cdots\mathbf{D}(\mathbf{W}_l^{a_i})^{\mathrm{T}}\cdots\mathbf{D}(\mathbf{W}_l^{b_1})^{\mathrm{T}} \cdot \partial\ell_s(\mathbf{W})/\partial g^j\|_2^2 \cdot \|h_{p-1}^{j,a_i}\|_2^2$$

$$(56)$$

From Lemma B.11 and Corollary B.9, with probability at least $1\text{-}\mathcal{O}(nN)\exp[-\Omega(m)]$, we have

$$\|\nabla_{\mathbf{W}_p^{a_i}}\ell_s(\mathbf{W})\|_2^2 \leq \sum_{j=i}^{N} \mathcal{O}(\|\partial\ell_s(\mathbf{W})/\partial g^j\|_2^2) \cdot \mathcal{O}(1) \cdot \|\mathbf{x}_t\|_2^2$$

$$= \sum_{j=i}^{N} \left( \mathcal{O}(\kappa_j^2\ell_s(\mathbf{W})) \cdot \mathcal{O}(1) \cdot \|\mathbf{x}_t\|_2^2 \right)$$

$$(57)$$

$$= \mathcal{O}\left( \ell_s(\mathbf{W}) \cdot \|\mathbf{x}_t\|_2^2 \cdot \sum_{j=i}^{N} \kappa_j^2 + c_3 \right)$$

where the constant $c_3$ can compensate for the estimation loss caused by all the approximation calculations in Eq. (55). Similarly, by Eq. (42), we can prove the situation while $q = b_i$, i.e., $\|\nabla_{\mathbf{W}_p^{b_i}}\ell_s(\mathbf{W})\|_2^2 \lesssim \mathcal{O}\left( \ell_s(\mathbf{W}) \cdot \|\mathbf{x}_t\|_2^2 \cdot \sum_{j=i}^{N} \kappa_j^2 + c_3 \right)$.

$$\square$$

## B.4 The proof of Theorem 3.4.

**Theorem 3.4.** For UNet in Eq. (16), assume $M_0 = \max\{\|b_i \circ a_i\|_2, 1 \leq i \leq N\}$ and $f_N$ is $L_0$-Lipschitz continuous. $c_0$ is a constant related to $M_0$ and $L_0$. Suppose $\mathbf{x}_t^{\epsilon_\delta}$ is an perturbated input of the vanilla input $\mathbf{x}_t$ with a small perturbation $\epsilon_\delta = \|\mathbf{x}_t^{\epsilon_\delta} - \mathbf{x}_t\|_2$. Then we have

$$\|\mathbf{UNet}(\mathbf{x}_t^{\epsilon_\delta}) - \mathbf{UNet}(\mathbf{x}_t)\|_2 \leq \epsilon_\delta \left[ \sum_{i=1}^{N} \kappa_i M_0^i + c_0 \right], \qquad (58)$$

where $\mathbf{x}_t = \sqrt{\bar{\alpha}_t}\mathbf{x}_0 + \sqrt{1 - \bar{\alpha}_t}\epsilon_t$, $\epsilon_t \sim \mathcal{N}(\mathbf{0}, \mathbf{I})$, $N$ is the number of the long skip connections.

*Proof.* From Eq. (16), we have

$$\mathbf{UNet}(\mathbf{x}_t) = f_0(\mathbf{x}_t) = b_1 \circ [\kappa_1 \cdot a_1 \circ \mathbf{x}_t + f_1(a_1 \circ \mathbf{x}_t)], \qquad (59)$$

and

$$\mathbf{UNet}(\mathbf{x}_t^{\epsilon_\delta}) = f_0(\mathbf{x}_t^{\epsilon_\delta}) = b_1 \circ [\kappa_1 \cdot a_1 \circ \mathbf{x}_t^{\epsilon_\delta} + f_1(a_1 \circ \mathbf{x}_t^{\epsilon_\delta})]. \qquad (60)$$

Using Taylor expansion, we have

$$\mathbf{UNet}(\mathbf{x}_t^{\epsilon_\delta}) - \mathbf{UNet}(\mathbf{x}_t) = b_1 \circ [\kappa_1 \cdot a_1 \circ (\mathbf{x}_t^{\epsilon_\delta} - \mathbf{x}_t) + f_1(a_1 \circ \mathbf{x}_t^{\epsilon_\delta}) - f_1(a_1 \circ \mathbf{x}_t)]$$

$$= b_1 \circ a_1 \circ [\kappa_1 \cdot (\mathbf{x}_t^{\epsilon_\delta} - \mathbf{x}_t) + (\mathbf{x}_t^{\epsilon_\delta} - \mathbf{x}_t)^T \nabla_z f_1(z)|_{z=a_1 \circ \mathbf{x}_t}].$$

$$(61)$$

Therefore,

$$\|\mathbf{UNet}(\mathbf{x}_t^{\epsilon_\delta}) - \mathbf{UNet}(\mathbf{x}_t)\|_2 = \|b_1 \circ a_1 \circ [\kappa_1 \cdot (\mathbf{x}_t^{\epsilon_\delta} - \mathbf{x}_t) + (\mathbf{x}_t^{\epsilon_\delta} - \mathbf{x}_t)^T \nabla_z f_1(z)|_{z=a_1 \circ \mathbf{x}_t}]\|_2$$

$$\leq \epsilon_\delta \cdot \|b_1 \circ a_1\|_2 \cdot (\kappa_1 + \|\nabla_z f_1(z)|_{z=a_1 \circ \mathbf{x}_t}\|_2)$$

$$(62)$$

Next, we estimate the $\|\nabla_z f_1(z)|_{z=a_1 \circ \mathbf{x}_t}\|_2$, From Eq. (16), we have

$$\nabla_z f_1(z) = \nabla_z b_2 \circ [\kappa_2 \cdot a_2 \circ z + f_2(a_2 \circ z)]$$
$$= \kappa_2 \cdot b_2 \circ a_2 \circ \mathbf{I} + b_2 \circ a_2 \circ \nabla_u f_2(u)\,|_{u=a_2\circ z}, \tag{63}$$

Therefore, we have

$$
\begin{aligned}
\|\nabla_z f_1(z)\,|_{z=a_1\circ\mathbf{x}_t}\|_2 &= \|(\kappa_2 \cdot b_2 \circ a_2 \circ \mathbf{I} + b_2 \circ a_2 \circ \nabla_u f_2(u)\,|_{u=a_2\circ z})|_{z=a_1\circ\mathbf{x}_t}\|_2 \\
&= \|\kappa_2 \cdot b_2 \circ a_2 \circ \mathbf{I} + b_2 \circ a_2 \circ \nabla_u f_2(u)\,|_{u=a_2\circ a_1\circ\mathbf{x}_t}\|_2 \\
&\le \|b_2 \circ a_2\|_2 \cdot (\kappa_2 + \|\nabla_u f_2(u)\,|_{u=a_2\circ a_1\circ\mathbf{x}_t}\|_2).
\end{aligned} \tag{64}
$$

From Eq. (62) and Eq. (64), let $\Delta = \|\mathbf{UNet}(\mathbf{x}_t^{\epsilon_\delta}) - \mathbf{UNet}(\mathbf{x}_t)\|_2$, we have

$$
\begin{aligned}
\Delta &\le \epsilon_\delta \cdot \|b_1 \circ a_1\|_2 \cdot (\kappa_1 + \|\nabla_z f_1(z)\,|_{z=a_1\circ\mathbf{x}_t}\|_2) \\
&\le \epsilon_\delta \cdot \|b_1 \circ a_1\|_2 \cdot [\kappa_1 + \|b_2 \circ a_2\|_2 \cdot (\kappa_2 + \|\nabla_u f_2(u)\,|_{u=a_2\circ a_1\circ\mathbf{x}_t}\|_2)] \\
&\le \epsilon_\delta \cdot (\kappa_1\|b_1 \circ a_1\|_2 + \kappa_2\|b_1 \circ a_1\|_2 \cdot \|b_2 \circ a_2\|_2 + \|b_1 \circ a_1\|_2 \cdot \|b_2 \circ a_2\|_2 \cdot \|\nabla_u f_2(u)\,|_{u=a_2\circ a_1\circ\mathbf{x}_t}\|_2) \\
&\le \epsilon_\delta \Big[\sum_{i=1}^{N}\big(\kappa_i \prod_{j=1}^{i}\|b_j \circ a_j\|_2\big) + \prod_{j=1}^{N}\|b_j \circ a_j\|_2 \cdot L_0\Big] \\
&\le \epsilon_\delta \cdot (\kappa_1 M_0 + \kappa_2 M_0^2 \ldots + \kappa_N M_0^N + c_0),
\end{aligned} \tag{65}
$$

where $c_0 = M_0^N L_0$, and hence we have $\|\mathbf{UNet}(\mathbf{x}_t^{\epsilon_\delta}) - \mathbf{UNet}(\mathbf{x}_t)\|_2 \le \epsilon_\delta \left[\sum_{i=1}^{N}\kappa_i M_0^i + c_0\right]$ for UNet in Eq. (16) and $\mathbf{x}_t = \sqrt{\bar{\alpha}_t}\mathbf{x}_0 + \sqrt{1-\bar{\alpha}_t}\epsilon_t$, $\epsilon_t \sim \mathcal{N}(\mathbf{0}, \mathbf{I})$, with small perturbation $\epsilon_\delta$.

$\square$

Next, we demonstrate through empirical experiments that $M_0$ in Theorem 3.4 is generally greater than 1. As shown in Fig. 11, we directly computed $\|b_i \circ a_i\|_2$ for the Eq. (16) on the CIFAR10 and CelebA datasets. We observe that $\|b_i \circ a_i\|_2$ exhibits a monotonically increasing trend with training iterations and is always greater than 1 at any iteration. Since $M_0 = \max \|b_i \circ a_i\|_2, 1 \le i \le N$, we conclude that $M_0 \ge 1$.

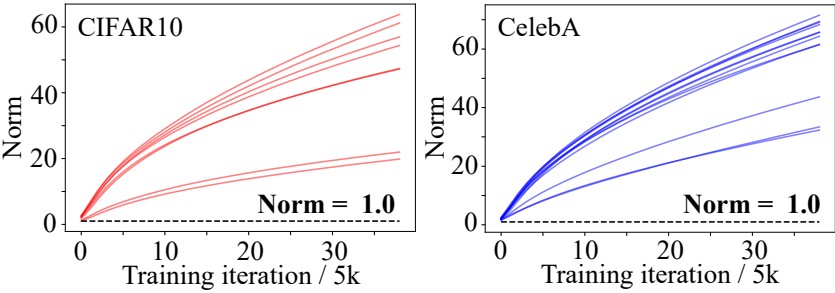

Figure 11: The norm of $\|b_i \circ a_i\|_2$ on different dataset.

# C  Other experimental observations

Fig. 12 is a supplement to Fig. 1 in the main text. We investigate the case of UNet with a depth of 12 on the CIFAR10 dataset, where each image represents the visualization results obtained with different random seeds. We observe that the original UNet consistently exhibits feature oscillation, while our proposed method effectively reduces the issue of oscillation, thereby stabilizing the training of the diffusion model.

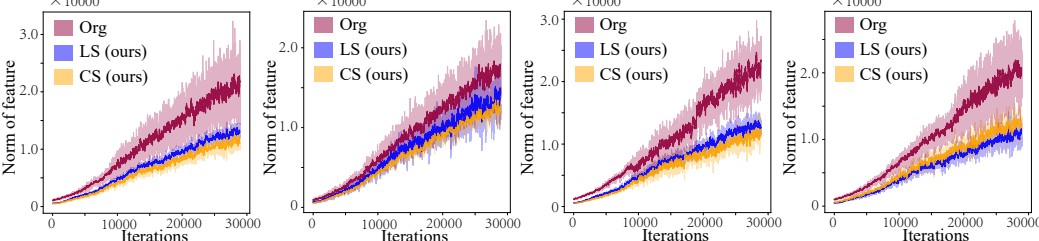

Figure 12: More examples for feature oscillation under different random seeds (UNet's depth = 12).

In the main text, we specify that the domain of $\kappa$ in CS should be (0,1). This ensures stable forward and backward propagation of the model and enhances the model's robustness to input noise, as discussed in Section 3. Otherwise, it would exacerbate the instability of the model training.

In Fig. 13, we illustrate the impact of having $\kappa > 1$ on the performance of the diffusion model. We observe that generally, having $\kappa > 1$ leads to a certain degree of performance degradation, especially in unstable training settings, such as when the batch size is small. For example, in the case of ImageNet64, when the batch size is reduced to 256, the model's training rapidly collapses when $\kappa > 1$. These experimental results regarding $\kappa$ align with the analysis presented in Section 3.

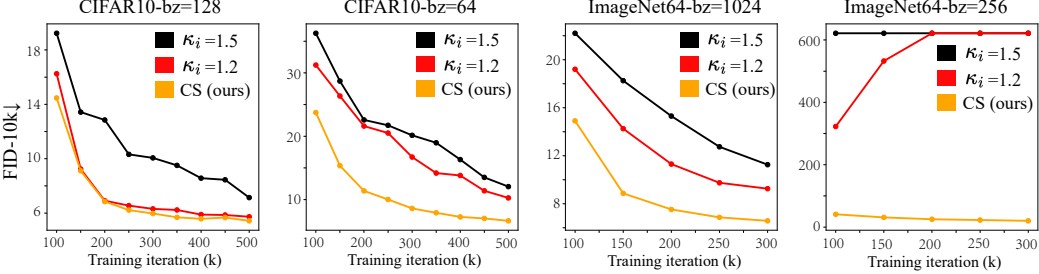

Figure 13: The situation that the $\kappa_i = \kappa > 1$ under different settings.

