# OpenReview forum: "ScaleLong: Towards More Stable Training of Diffusion Model via Scaling Network Long Skip Connection"
_NeurIPS.cc/2023/Conference — NeurIPS 2023 poster_

### Official Review · Reviewer_Ehww · 2023-07-07

**Soundness:** 3 good
**Presentation:** 3 good
**Contribution:** 3 good
**Rating:** 6
**Confidence:** 2

**Summary:**

They state that diffusion models using Unet suffer from unstable training
and oscillations of features and gradients. They also state that while this is sensitive to coefficients related to scaling the skip connections of Unet. They set out to provide an explanation and more robust scaling methods for the skip connections to address this. Their methods are constant scaling based exponentially on the depth of the skip connection, and learnable scaling. The methods seem to consistently reduce FID versus the same model with regular scaling.

**Strengths:**

I believe this paper’s contribution is simple and meaningful, and written
clearly. The experiments seem thorough to changes made and the questions you would have reading it. I cannot fully speak either way to novelty in this area, but it seems like a sound paper.


**Weaknesses:**

I did not see meaningful weaknesses in the paper.

**Questions:**

N/A

---

> ### Author Rebuttal · Authors · 2023-08-10
>
> **`(1)` Thank you for your encouragement and taking the time to review our article.** We will further improve our paper based on the suggestions of other reviewers.

---

> > ### Comment · Reviewer_Ehww · 2023-08-17
> > **Rebuttal read**
> >
> > I confirm I have read the rebuttal and would like to keep my score.

---

### Official Review · Reviewer_VLJa · 2023-07-07

**Soundness:** 3 good
**Presentation:** 2 fair
**Contribution:** 3 good
**Rating:** 6
**Confidence:** 3

**Summary:**

U-Net is the most popular neural network backbone for diffusion models. In U-Net, the long skip connection (LSC) links the long-distant information near to the input and the intermediate network outputs. However, they suffers from unstable training, which is resolved by scaling down the LSC coefficients. This paper addresses the theoretical aspects of stabilizing LSC, robustifies the training process, and accelerates this. Both hand-crafted and learning-based parameterization are proposed.

**Strengths:**

* The organization of the paper is clear, and the figures of motivating experiments are well demonstrated.
* The method and simple yet efficient, improving both the training speed and sampling performance while requiring only some hand-crafted hyperparameters (CS) or lightweight learning module (LS).

**Weaknesses:**

* This work does not deal with more recent diffusion model baselines such as EDMs, which is much more powerful than the methods that are compared in this work.
* Prior, related works are not introduced, especially for those who are not familiar with the scaling of skip connections in other network architectures.

**Questions:**

* This work focuses on scaling the skip connection coefficients U-Net-based diffusion models. Does this technique also hold for other U-Net based methods rather than diffusion models?
* Can you add some figures or tables that addresses the optimal value of $\kappa$ in CS cases, like the $\kappa$ from LS in Figure 7?
* Please write the full name of IE and SE in line 330.
* Please address the number of sampling steps in each methods.
* Does parameterizing LC to some additional module (Figure 4) and training the network end-to-end adequately learn $\kappa_i$?

===========

Corrections
* Line 85: serir --> series
* Line 86: UNettraining --> UNet training

**Limitations:**

The authors adequately addressed the limitations and introduced further directions.

---

> ### Author Rebuttal · Authors · 2023-08-10
>
> Thank you for the insightful and positive comments! In the following, we provide our point-by-point response and hope our response helps address your concerns. We also look forward to the subsequent discussion which may further help solve the current issues.
>
> **`(1)` About the baseline EDMs.**  Thank you for your suggestion. Since our method focuses on the neural network architecture for the diffusion model,  it could be applicable to EDMs which also uses UNet for training. However, this needs to be thoughtfully tested. Since EDMs require  much longer training time, such as 16 GPU days for CIFAR10 and over 300 GPU days for ImageNet64,  it is really challenging for us to finish the experiments within the rebuttal phase. In the revision, we will try our best to evaluate our method on EDMs and report the results in revision.
>
> **`(2)` About the related works.** Thank you for your suggestion.  At present, due to the limited space (9 pages), we have to briefly introduce the related works in Line 22-26 and Line 101-128, and spend much space to introduce three theoretical analysis from three different aspects, and our proposed methods. We very much appreciate your suggestions about   introducing more related works, including the  previous scaling methods of skip connections, and will try to discuss more in the revision since  the final version often allows us to use an extra page.
>
> **`(3)` Our technique may not directly improve other UNet based tasks.** Our analysis is based on diffusion models. For example, our mathematical derivation relies on the particularity of the diffusion model, such as the approximately normal distribution of the network's predictions (i.e., noise). For other typical UNet-based scenarios, such as image segmentation and depth prediction, the neural network's output consists of segmentation masks or depth maps which may not fully satisfy our analysis needs.
>
> While our theories cannot be directly applied to other tasks, our technology, especially the experimental setup based on our theories, may not be directly transferable to other UNet-based scenarios. In the future, we will test our proposed LS and CS in some common UNet-based scenarios, such as image segmentation, to further enhance the versatility of our approach.
>
> **`(4)` The optimal value of $\kappa$ in CS cases** please refer to rebuttal PDF Fig.1 (c). The values of $\kappa$ in CS are 0.5 (for CIFAR10), 0.5 (for CelebA), 0.8 (for MS-COCO) and 0.95 (for ImageNet64).
>
>
> **`(5)` About the full name of IE and SE**. Thanks for your suggestion. We will add the full name of IE (Instance enhancement batch normalization [1]) and SE (Squeeze-and-excitation networks [2]) in the revision.
>
>
> **`(6)` About the sampling steps in each methods**. We follow the default and official sampling settings in UViT. Specifically, for CIFAR10 and CelebA64, the experiments used the "Euler-maruyama-sde" method with 1000 sampling steps for sampling. On the other hand, for MS-COCO and ImageNet64, the experiments used the "DPM solver" method with 50 sampling steps.
>
> **`(7)` About the LC parameterizing**. Apologies, we may not fully understand your question. Let me try to clarify:
>
> - If you're asking whether LS can be used to end-to-end adaptively learn $\kappa_i$, then according to the explanation in Section 4.2, our LS is originally designed for adaptive learning of $\kappa_i$, rather than being a fixed value like CS.
>
> - If you're inquiring about the feasibility of applying LS with additional (other) modules for adaptive learning of $\kappa_i$, Table 2 explores the use of alternative modules for learning $\kappa_i$. However, the results indicate that the current structure already achieves satisfactory performance.
>
> I hope my response addresses your issue. If not, please let me know.
>
>
> **`(8)` For the typos**,  per your suggestions, we will fix them in revision.
>
> [1] Senwei et al. Instance enhancement batch normalization: an adaptive regulator of batch noise. AAAI 2020.
>
> [2] Jie Hu et al. Squeeze-and-excitation networks. TPAMI 2017.

---

> > ### Comment · Reviewer_VLJa · 2023-08-17
> > **Response to the rebuttal**
> >
> > Thank you for the response. For the LC parameterization, my question was the latter: for some cases, the CS case adequately achieves good performance for learning $\kappa_i$, and is it beneficial to additively learn with some adaptive module?
> >
> > The rebuttal resolved most of the questions. As I expect this work to contribute to the architectural design of the well-performing diffusion model, I am raising the review score.

---

### Official Review · Reviewer_xZoY · 2023-07-07

**Soundness:** 3 good
**Presentation:** 3 good
**Contribution:** 3 good
**Rating:** 7
**Confidence:** 3

**Summary:**

This paper proposes to scale the skip connection in diffusion model Unets by an exponential factor. The authors should that the feature norms of vanilla Unets oscillate across batches, and that their proposed method results in much smaller feature output oscillations. They conjecture that this method stabilizes training, and prove bounds on the oscillation which their method decreases. The proposed method shows faster convergence and shows improvements across multiple models, datasets, model sizes.

**Strengths:**

1. The authors method follows prior work on scaling block output and/or the skip connection, where such scaling methods have been shown to increase stability/robustness and result in Lipschitz continuous models.
1. The learnt scales in LS method seems to somewhat mirror the author's theoretical exponential CS method, providing further validation.
1. The authors method results in faster convergence and shows improvements across multiple models, datasets, model sizes, and compares favourably to some other scaling methods.
1. The author's proposed method can be very easily adapted to existing models.

**Weaknesses:**

1. The authors show that feature norms oscillate across samples, and they say this implies parameters must also be oscillating (line 47, line 172). While the authors demonstrate that feature norms do oscillate (and I am willing to agree the same may be true of gradient norms), this may not directly cause the parameters/updates to oscillate.
1. The authors works (and proposed exponential CS solution) is remarkably similar to [1], where a scaling of $b^l$ is proposed for the block output (compared to this paper's scaling of the skip connection by a similar value). An experimental comparison with this method should be added to Table 3, or perhaps the similarities/differences discussed.
1. Discussion of prior related works should be expanded. For example, [2] proposes scaling of the skip connection and/or the block output, and has similar exponential bounds.
1. The constants in theorem 3.1 are ignored (assumed to be 1) in line 270 to derive the values for the scaling parameter. If this constant is for example 10 or 0.1, that will dramatically change the proposed values for CS. Further investigation into this constant is required.


[1] Hanin, B., & Rolnick, D. (2018). How to Start Training: The Effect of Initialization and Architecture. Advances in Neural Information Processing Systems, 31, 571–581.
[2] Balduzzi, D., Frean, M., Leary, L., Lewis, J.P., Ma, K.W. &amp; McWilliams, B.. (2017). The Shattered Gradients Problem: If resnets are the answer, then what is the question?. <i>Proceedings of the 34th International Conference on Machine Learning</i>, in <i>Proceedings of Machine Learning Research</i> 70:342-350 Available from https://proceedings.mlr.press/v70/balduzzi17b.html.

**Questions:**

1. The authors derive bounds for oscillations (eq 4), and they show that their methods decreases this bound, but how tight is this bound? Comparing the minimum and maximum norms predicted by this formula, with that in Figure 1., is important to show if this bound is indeed useful, or perhaps their proposed method has some other side effect that results in the performance improvements.
1. Direct evidence of parameter oscillation (mentioned in line 47, 172 can be shown), perhaps by comparing a per-parameter Adam style "m" and "v" across the baseline method and the author's method. If oscillation of parameters is indeed smaller, one would perhaps expect the norm of "m/v" to be larger in the author's case. Is this indeed the case?
1. Given that the exponential change in scaling with increasing total number of layers, does the performance of the method suffer compared to $\frac{1}{\sqrt{2}}-CS$ as number of layers increases to more depths (as large depth will cause the skip connection to essentially be scaled to zero)?

Minor presentation issues (authors are not expected to respond to these, as they are minor issues that can be fixed later):
1. Line 85 "serir" should perhaps be "series"?
1. In table 1, comparisons to author's method "x+LS/CS" should perhaps be placed directly below "x", to make it easier to compare scores.
1. The values for CS for table 1 are mentioned in the supplementary, but perhaps they should be mentioned at line 272 or in 315.
1. Quotes around 'CS.py' in supplementary above Figure 2 are inverted.
1. $\frac{1}{\sqrt{2}}-CS$ reads as "1 by square root 2 minus CS", and for a moment I was trying to think what $CS$ are the authors subtracting from  $\frac{1}{\sqrt{2}}$. Perhaps a better notation for this may be something along the lines of $(CS)\frac{1}{\sqrt{2}}$?

**Limitations:**

-

---

> ### Author Rebuttal · Authors · 2023-08-10
>
> Thank you for the insightful and positive comments! In the following, we provide our point-by-point response and hope our response helps address your concerns. We also look forward to the subsequent discussion which may further help solve the current issues.
>
> **`(1)` For parameter oscillation,**  we indeed observe it as discussion below, and our approach effectively mitigates this oscillation.
>
>  As shown in Fig.2 in the attached PDF, taking UViT on CIFAR-10  as an example, we select several parameters from both the encoder and decoder's final layers. We then computed the average sliding variance (using a window size of 5) of these parameters during training for the three methods (org, LS, CS). This approach provides more direct observation of parameter oscillation compared to monitoring the norm of "m/v." From the visualization results, one can observe that parameter oscillation does occur, and our method effectively mitigates this oscillation to stabilize model training.
>
> **`(2)` For the scaling method  on the block output in the work [1],** we compare it with ours in  Table 2 in the attached PDF, where the scaling method in [1] is denoted as $b^l$. We follow three settings from  the work [1]  by respectively setting $b=0.90, 0.75, 0.50$, and find that this kind of scaling method on the block output cannot achieve good  performance improvement.
>  Therefore, for stable training of the diffusion model, we believe that applying scaling on long skip connections, similar to our proposed CS, would be more effective. We will include this comparison into our revision.
>
> **`(3)` In the revision, we will discuss more related works.** For the work [2], our work mainly differs from it in two notable differences.
>
> - (**Architecture**) Our method focuses on the UNet architecture, particularly the long skip connections, which are not considered in [2]. Most of the discussions and analyses about ResNet (Kaiming He 2016) from [2] cannot be directly applied to UNet.
>
> - (**Scaling location**) The work [2] also primarily focuses on scaling at the block output, rather than on scaling at skip connections. As mentioned in section 3.3 of paper [2], scaling at skip connections may not work well in practice for ResNet (Kaiming He 2016). On the contrary, in the scenario of UNet and diffusion models, we notice that scaling at long skip connections is more effective than scaling at the block output.
>
> Our work's analysis and conclusions contribute to an expanded understanding of scaling strategies in neural network architectures within the community. In the revision, we will include the above discussion to further strengthen our work.
>
> **`(4)` For the constant $c_1$ in theorem 3.1,**
> as mentioned in Line 270-272, we use a rough estimate for constant $c_1 \approx 1$ which is valid in many experiments.
>  Moreover, per your suggestion,  we conduct statistical analysis on different datasets such as CIFAR10, CelebA, and ImageNet64, and find that the real range of the constant $c_1$ is  $(0.752, 1.060)$ for analysis in Line 270-272, which is close to our rough estimate $c_1\approx 1$ and will not dramatically change the proposed values for CS.  In the revision, we will modify this section to enhance our writing.
>
> **`(5)` For the oscillation bound in Eq (4),**
> due to the use of some approximations and consideration of worst-case scenarios in Eq (4), our bound may not be very tight. And it's also not easy to be visualized in Figure 1 since some parameters are hard to measure during training directly and quantitatively, e.g., $c_2$. In this paper, Eq(4) serves as a qualitative estimation of feature oscillations, providing inspiration and explanation of our proposed scaling method from a feature map perspective. This indicates that the estimation of Eq(4) is meaningful and insightful.
> Moreover, we are the first to conduct such an analysis on the diffusion model of the UNet neural network architecture. In future work, we will attempt more sophisticated analytical approaches to obtain tighter bounds.
>
>
> **`(6)` The comparison between    $\frac{1}{\sqrt{2}}$-CS and ours on very deep network.**  If the number of layers is large enough, indeed, our scaling will tend to zero. However, this situation is relatively rare in the practical diffusion model tasks using UNet. On the most popular image or text generation tasks, a UNet usually doesn't have a large number of long skip connections. Therefore, in this practical real-world setting,  our scaling factor will not be scaled to zero, and our methods are more effective than $\frac{1}{\sqrt{2}}$-CS.
>
> As technology advances, our proposed scaling strategy will need further refinement to handle potential cases where diffusion models may be employed in extremely deep networks. Additionally, if a long skip connection approaches zero but the performance can be maintained, it may not necessarily be a bad thing, as these connections can be almost removed to save memory and improve model inference speed without sacrificing performance.
>
>
> **`(7)` For the minor presentation issues**,  per your suggestions, we will fix them in revision.
>
> [1] Hanin, B. et al. How to Start Training: The Effect of Initialization and Architecture. NeurIPS 2018.
>
> [2] Balduzzi, D. et al. The Shattered Gradients Problem: If resnets are the answer, then what is the question? ICML 2017.

---

> > ### Comment · Reviewer_xZoY · 2023-08-17
> >
> > Thank you for the detailed response.
> >
> > In particular, I am satisfied with your empirical validation of constant C1, and Table 2 comparisons to prior works. The charts in Figure 2 of the global author rebuttal are also welcome. The figure does indeed show the "oscillations" the author's hypothesize and try to fix by bounding.
> >
> > As such, I am raising my review score.

---

### Official Review · Reviewer_Ttzr · 2023-07-13

**Soundness:** 3 good
**Presentation:** 3 good
**Contribution:** 3 good
**Rating:** 5
**Confidence:** 3

**Summary:**

In this paper, the authors focus on the challenge of the instability arising from the commonly adapted U-Net architecture for diffusion models. In particular, the authors start by theoretically analyzing the influence of the coefficients of long skip connects in U-Net-based diffusion models, specifically on the stability of the forward and backward propagation, along with the robustness of the network. Motivated by the theoretical discussions, the authors propose two corresponding scaling methods that adapt the coefficients of the long skip connections for more stable training, including a constant scaling method and a learnable scaling method, which is straightforward and ideally similar to that in dynamic neural networks and meta learning, but backed by solid theoretical analysis. Experiments on four datasets validate the effectiveness of the proposed two forms of scaling manners, with key codes provided in the supplement.

**Strengths:**

- The proposed constant and learnable scaling methods are very straightforward, but with solid theoretical analysis and guarantee. The results shown in Fig. 1 are also promising that successfully alleviate the oscillations.
- Extensive ablation studies including the robustness of LS to network architecture are provided.
- Key codes are provided in the supplementary material for handy reproducibility.


**Weaknesses:**

- Method: The second part of learnable scaling method in this paper is algorithmically related to meta learning and dynamic neural networks. However, there is a lack of discussions on related works. Also, it would be great if the authors could discuss some alternative ways for learnable scaling by borrowing the ideas from meta learning and dynamic neural networks.
- Experiments: The authors are encouraged to provide some qualitative comparative results with existing methods, as a complement to the quantitative results reported at, for example, Tab. 1.

The instability issue in diffusion models with U-Net is not my primary research area. As such, I may not be capable of correctly evaluating the novelty of this paper. I will refer to the comments of other reviewers for my final justification.


**Questions:**

Please kindly refer to the weakness section for more details.

**Limitations:**

Limitations are discussed in Sect. 6 of the paper.

---

> ### Author Rebuttal · Authors · 2023-08-10
>
> Thank you for the insightful and very positive comments! In the following, we provide our point-by-point response and hope our response helps address your concerns. We also look forward to the subsequent discussion which may further help solve the current issues.
>
> **`(1)` Per your suggestion, here we discuss some alternative ways for learnable scaling** by borrowing the ideas from meta-learning and dynamic neural networks.
>
> - For dynamic neural networks, the modules like LS can be viewed as attention modules. The work [1] suggests that such modules can be considered as adaptive feature-regulating dynamic neural networks. So by replacing or designing improved attention modules, we can effectively borrow ideas from dynamic neural networks to enhance the performance of diffusion models. Indeed, in our manuscript, we have already explored different attention modules. For example, Table 2 in the main text also presents the performance of some other alternative modules.
>
> - For meta-learning. We can loosely follow the steps inspired by MAML [2]. First, We can regard the scaling module $\mathcal{M}$ as a meta-learner in meta-learning, and regard each sample $\mathcal{T}_ i$  as a task. For the inner loop, given a task (sample) $\mathcal{T}_ i$, the vanilla diffusion loss can be used to tune $\mathcal{M}$  via a step of gradient descent to adapt $\mathcal{M}$  to the task $\mathcal{T} _ i$  and obtain a new task-specific scaling module $\mathcal{M}_ {T_ i}$.  In the outer loop, we feed the sample $\mathcal{T}_ i$ into the diffusion model with a task-specific scaling module $\mathcal{M}_ {T_ i}$ again, and update the diffusion model and the scaling module $\mathcal{M}$ via gradient. The benefits of this meta-learning is that we can learn a meta model $\mathcal{M}$ as the scaling module, and  adapt it to a specific sample for generating  LSC weights. Though this idea is promising,this meta-learning method requires to compute the Hessian matrix, and thus suffers from high computational cost which is not very suitable for diffusion models whose model size is often large.
>
>    In the revision, we will supplement the above discussion, and try to provide further validation experiments and analysis.
>
> [1] Han Y et al. Dynamic neural networks: A survey[J]. TPAMI, 2021
>
> [2] Finn C et al. Model-agnostic meta-learning for fast adaptation of deep networks[C]. ICML 2017.
>
> **`(2)` For qualitative comparative results,**  we have already provided some visualizations of our method (in Tab. 1) in Appendix for qualitative analysis. From the visual results (namely, the generated images), our methods can generate highly qualified images. Per your suggestion, we will include more comparisons with other methods in the revision.

---

> > ### Comment · Area_Chair_hSdX · 2023-08-18
> > **Reviewer Ttzr**
> >
> > Dear Reviewer Ttzr,
> >
> > Could you please comment on whether the rebuttal addresses your comments and concerns?
> >
> > Best,
> > AC

---

### Official Review · Reviewer_7ePs · 2023-07-25

**Soundness:** 2 fair
**Presentation:** 2 fair
**Contribution:** 2 fair
**Rating:** 6
**Confidence:** 2

**Summary:**

The paper discusses the stability issues observed while training UNet in diffusion models, and theorizes on the role of Long Skip Connections (LSCs) in causing this instability.

Diffusion models (DMs), lauded for their ability to model realistic data distributions, involve a forward and a reverse diffusion process.

Most DMs use UNet as their backbone due to its use of LSCs which facilitate long-distance information aggregation and prevent vanishing gradient issues.

However, despite the use of shared UNet for predicting injected noise at each step, instability is noticed during training.

The paper's main contributions revolve around investigating this instability and deriving effective (and efficient) methods to address it.

Specifically, it is theoretically proven that the coefficients of LSCs in UNet significantly affect the stability of forward and backward propagation as well as the robustness of UNet.

The paper also proposes two coefficient scaling methods, Constant Scaling (CS) and Learnable Scaling (LS), designed to adjust the coefficients of LSCs in UNet for training stability.

CS involves setting the coefficients as a series of exponentially-decaying constants while LS uses a small shared network to predict the scaling coefficients for each LSC.

**Strengths:**

- One of the key strengths of the paper is its rigorous theoretical analysis. It provides a comprehensive understanding of the instability of UNet in diffusion models by focusing on the significant impact of the coefficients of Long Skip Connections (LSCs). It's an important insight that broadens the understanding of UNet's performance in diffusion models.


- The paper proposes two novel scaling methods, Constant Scaling (CS) and Learnable Scaling (LS), designed to enhance the stability of the UNet training process. These simple methods address the identified problem and provide tangible ways to improve training stability, indicating a proactive approach to problem-solving.



**Weaknesses:**

- The paper primarily focuses on the role of Long Skip Connections (LSCs) in causing training instability in UNet. While LSCs are a significant part of the model, the authors might have explored other potential contributing factors to this instability as well. Broadening the scope of their investigation could potentially have led to a more comprehensive understanding of the issue.


- The placement of Figure 1 doesn't align well with its corresponding description, causing some disconnection. Ideally, figures on the first page should offer readers a quick, intuitive understanding of the paper's content. In its present state, Figure 1 seems to lack sufficient information to accomplish this. If this positioning is due to space constraints, please consider relocating it to a more suitable page.



**Questions:**

-

**Limitations:**

-

---

> ### Author Rebuttal · Authors · 2023-08-10
>
> Thank you for your insightful and positive comments.  In the following, we provide our point-by-point response and hope our response helps address your concerns. We also look forward to the subsequent discussion which may further help solve the current issues.
>
>
> **`(1)` In addition to LSCs, we  have also explored other potential contributing factors.** We found that the decoder of UNet is also a significant part of the model. We have extensively explored how to apply scaling strategy  to the output of each block in the decoder, hoping to stabilize model training. However, we find that this approach cannot achieve satisfactory performance for diffusion models, and are exploring the reasons behind.
>
>
>
> **`(2)` For the layout of the figures and the corresponding text**,  per your suggestion, we will arrange them in the revision so that the figures and the corresponding text are close,  improving the readability.

---

> > ### Comment · Reviewer_7ePs · 2023-08-16
> >
> > I appreciate the authors' thoughtful response.
> > I've carefully read other reviews and the authors' rebuttals, and I've decided to stick with my original score.
> > Based on the feedbacks (particularly concerning typos, figures, and writing), I believe the work can be further polished.

---

### Official Review · Reviewer_KfUJ · 2023-07-26

**Soundness:** 3 good
**Presentation:** 4 excellent
**Contribution:** 3 good
**Rating:** 7
**Confidence:** 2

**Summary:**

The paper presents a study and algorithm for scaling UNet's long-range connections such that convergence and stability can be improved. The results are strong in the setup of training diffusion models.

**Strengths:**

- The paper is very well written. The text and graphs are polished and it's easy to follow the idea and contributions.
- The proposed modification of using an exponential scale over depth is simple and effective, which could be directly applicable to any UNet architecture.
- The experiments are thorough and the results are strong. The proposed CS and LS algorithms are shown to be better than the vanilla UNet and heuristic $1/\sqrt(2)$ scaling rule in terms of stability and performance.

**Weaknesses:**

- Explanation of the algorithm: Although the CS and LS algorithms are shown effective, an intuitive explanation on why they work is missing.
    - For CS, how is the direction of applying the exponentially decayed scale determined? For instance, can we do $\kappa_i=\kappa^{D-i-1}$? In the explanation L251-259, I don't see anything specific to the direction or order.
    - For LS,  I'm assuming there is no additional supervision to the calibration network. In that case, is there any explanation of why it can discover a reasonable scaling rule (Fig.7)? Since both the weights and the scale contribute to the final output of each block, I think decoupling these would be challenging without additional regularization and supervision. Is there any explanation why LS discovers a decaying scaling curve similar to the CS heuristics, but not an increasing scale?
- Experiments
    - For most of the experiments, the results are reported for "Org" (no scaling) and "CS-$1/\sqrt(2)$". However, Fig. 5 does not contain results for "CS-$1/\sqrt(2)$". How does the proposed method compare to it in terms of convergence?
- Significance
    - Although the paper primarily focuses on training diffusion models, many of the analyses seem general enough to be applicable to other scenarios that use UNets, e.g., depth prediction and segmentations. I wonder if this is true and if there is a reason for only showing results on the generation tasks.

**Questions:**

Minor:
- L86: There should be a space between Unet and training.
- Fig.3 (a): What does the unit m mean on the x-axis? I'm also missing the point this figure is trying to make, as referred to by L161-L163.


**Limitations:**

Yes

---

> ### Author Rebuttal · Authors · 2023-08-10
>
> Thank you for the insightful and very positive comments! In the following, we provide our point-by-point response and hope our response helps address your concerns. We also look forward to the subsequent discussion which may further help solve the current issues.
>
> **`(1)` For the direction of  the exponentially decayed scale in CS**, it is derived by our theory. Specifically,  if we use the reverse direction, namely,  $\kappa_i = \kappa^{N-i+1} \ (\kappa<1)$, the stability bound in Theorem 3 is extremely large.  The main  term of stability bound in Theorem 3  can be written as
> $\mathcal{S}_{\text{r}} =\sum_{i=1}^N\kappa_i M_0^i =  \kappa^NM_0 + \kappa^{N-1}M_0^2+...+ \kappa M_0^N$, and could be very large,  since $M_0^N$ is large when $N$ is large and scaling it by a factor $\kappa$ could not sufficiently control its magnitude (here $M_0>1$, please see Line 230 in manuscript).
>
> In contrast,  our default setting $\kappa_i=\kappa^{i-1}$ of CS  can well control the main  term in stability bound:
> $\mathcal{S} = \sum_{i=1}^N\kappa_i M_0^i =  M_0 + \kappa^{1}M_0^2+...+ \kappa ^{N-1}M_0^N$,
> where the larger  terms $M_0^{i+1}$ are weighted by smaller coefficients $\kappa^{i}$. In this way, $\mathcal{S} $ is much smaller than $\mathcal{S}_{\text{r}}$,  which shows the advantages of our default setting.
>
> Besides,  the following Table 1  also  compares the above two settings by using  UViT on Cifar10  (batch size = 64), and shows that our default setting exhibits significant advantages.
>
> | Training step                  | 5k    | 10k   | 15k   | 20k   | 25k   | 30k   | 35k   | 40k   | 45k   |
> |--------------------------------|-------|-------|-------|-------|-------|-------|-------|-------|-------|
> | $\kappa_i = \kappa^{N-i+1}$    | 67.26 | 33.93 | 22.78 | 16.91 | 15.01 | 14.01 | 12.96 | 12.34 | 12.26 |
> | $\kappa_i=\kappa^{i-1}$ (ours) | 85.19 | 23.74 | 15.36 | 11.38 | 10.02 | 8.61  | 7.92  | 7.27  | 6.65  |
>
>
> **`(2)` There are two possible reasons  for why  LS discovers a decaying scaling curve similar to the CS.**
> - On one hand, from a theoretical view, as discussed in our reply (1), for the $i$-th long skip connection $(1\leq i \leq N)$, the learnable $\kappa_i$ should be smaller to control the magnitude of $M_0^i$ better so that the stability bound, e.g. in Theorem 3, is small. This directly yields the decaying scaling strategy which is also learnt by the scaling network.
>
> - On the other hand, we can also analyze this observation in a more intuitive manner. Specifically, considering the UNet architecture, the gradient that travels through the $i$-th long skip connection during the backpropagation process influences the updates of both the first $i$ blocks in the encoder and the last $i$ blocks in the UNet decoder. As a result, to ensure stable network training, it's advisable to slightly scale the gradients on the long skip connections involving more blocks (i.e., those with larger $i$ values) to prevent any potential issues with gradient explosion.
>
>
>
> **`(3)` For the convergence curve of $1/\sqrt{2}$-CS**, we do not include it in Fig. 5 for more clear comparison, since too many curves may lead to unclear comparison and possible misunderstanding. But in Figure 1 of the vanilla manuscript, we have compared with $1/\sqrt{2}$-CS, and find our methods can better stabilize the UNet training. Moreover, in Table 1, our methods also show better synthesis performance than $1/\sqrt{2}$-CS. Per your suggestion, we have compared with $1/\sqrt{2}$-CS in terms of convergence.   Taking the setting about UViT on CIFAR10 as examples, Fig.1 (a) and (b) in the rebuttal PDF  show that our methods reveal much faster convergence behaviors than $1/\sqrt{2}$-CS.
>
>
> **`(4)` Our analysis are based on diffusion models, and may not be directly applicable to other applications**. For example, our mathematical derivation relies on the particularity of the diffusion model, such as the approximately normal distribution of the network's predictions (i.e., noise). For other typical scenarios, such as image segmentation and depth prediction, the neural network's output consists of segmentation masks or depth maps which do not fully satisfy our analysis needs. While our theories cannot be directly applied to other tasks, there is potential to bridge these gaps through more calibrated analyses. This could be an important direction for future work.
>
>
> **`(5)` For the value of $m$ in Fig.3 (a)**, it denotes the feature dimension of $\mathbf{x}_t$ (Line 158). The small subfigure in Fig.3(a) is to experimentally verify the distribution of $\mathbf{x}_0$ required in Lemma 3.2. For the larger subfigure, it is used to validate the conclusion of Lemma 3.2 that $\mathbb{E}(||\mathbf{x}_t||_2^2)$ is of the order of $\mathcal{O}(m)$. In conclusion, the theoretical analysis and findings in Lemma 3.2 are consistent with the real statistical results on three datasets (CIFAR, CelebA, and ImageNet) in Fig.3.
>
>
> **`(6)` For the typos**,  per your suggestions, we will fix them in revision.

---

> > ### Comment · Reviewer_KfUJ · 2023-08-18
> >
> > Thank you for the clear explanation. I'd like to keep my original rating of acceptance.

---

### Author Rebuttal · Authors · 2023-08-10

We provide the necessary charts for the rebuttal stage in the attached PDF. Reviewers are kindly requested to refer to them.

---

### Decision · Program_Chairs · 2023-09-21

**Decision:**

Accept (poster)

**Comment:**

This paper proposes two scaling methods to stabilize the training of UNets in Diffusion Models, and provides both theoretical and empirical support. The paper received unanimous accept recommendations (2 accepts, 3 weak accepts, and 1 borderline accept) from reviewers. The paper's strengths include a simple and effective approach, rigorous theoretical analysis, and thorough experimental analysis and good results. Most of the initial weaknesses surrounding discussion of related work, questions regarding the approach, and experiments were addressed by the rebuttal. The ACs agree with the reviewers that this paper can make a good contribution to the machine learning and computer vision communities; in particular, the proposed approach is simple and effective, and has potential to be applicable to various UNet architectures. The ACs recommend acceptance.